# Economic viability requires higher recycling rates for imported plastic waste than expected

Kai Li [1] ✉, Hauke Ward[1], Hai Xiang Lin[1,2] & Arnold Tukker [1,3]

The environmental impact of traded plastic waste hinges on how it is treated. Existing studies often use domestic or scenario-based recycling rates for imported plastic waste, which is problematic due to differences in recyclability and the fact that importers pay for it. We estimate the minimum required recycling rate (*RRR*) needed to break even financially by analysing import prices, recycling costs, and the value of recycled plastics across 22 leading importing countries and four plastic waste types during 2013–2022. Here we show that at least 63% of imported plastic waste must be recycled, surpassing the average domestic recycling rate of 23% by 40 percentage points. This discrepancy suggests that recycled plastics volumes from the global North-to-South trade may be underestimated. The country-specific *RRR* provided could enhance research and policy efforts to better quantify and mitigate the environmental impact of plastic waste trade.

Over the past decades, increasing globalisation has fragmented supply chains, making the assessment of life-cycle environmental impacts more challenging[1–3]. A similar trend has emerged in waste management. Since 2019, traded waste plastics have amounted to approximately five million tons per year[4]. Typically, this waste is exported from high-income countries to low-income countries, where labour and treatment costs are lower[5]. However, such exports to the Global South have raised major concerns due to potential mismanagement, which can have severe negative impacts on the environment, ecosystems, and human health[5–8]. Mismanaged plastic waste contributes to river pollution and is a significant factor in the 'plastic soup' found in oceans[9].

Recent publications indicate that globally less than 10% of waste plastics are recycled[5]. A significant amount of plastic waste is mismanaged in countries with underdeveloped waste collection and treatment systems[10]. For example, over half of the plastic waste in Indonesia is incinerated without recovering energy, and 5% is disposed of in uncontrolled dumpsites[11]. Evidence shows that more than 60% of marine litter plastics emissions annually come from the Philippines, India, Malaysia, and Indonesia[7]. Much of this plastic waste treated in

the Global South originates from the Global North, contributing to environmental plastic waste emissions[12].

In response to these concerns, China, a major importer of plastic waste, implemented a plastic import ban in 2018[8]. This decision redirected plastic waste exports to other countries, notably Malaysia, Indonesia, Turkey and Vietnam. To address potential negative impacts and prevent mismanagement abroad, the European Union (EU) has recently considered a ban on plastic waste exports to non-OECD countries[13], adhering to the principle that countries should be responsible for the proper treatment of their own waste[14].

While the global plastic recycling rate remains low, there is an implicit assumption that traded plastic is primarily recycled[4]. However, accurately determining the recycling rate for imported plastic waste in receiving countries is challenging due to measurement difficulties. Existing studies often rely on assumed domestic or scenario-based recycling rates, which lack robust data support. For example, Wen et al. quantified the changes in environmental impacts resulting from the shift of plastic waste imports from China to Southeast Asia, using assumed domestic recycling rates of 10 to 40% for five Southeast Asian countries[8]. Similarly, Bourtsalas et al. estimated the environmental

[1]Institute of Environmental Sciences (CML), Leiden University, Leiden, The Netherlands. [2]Delft Institute of Applied Mathematics, Delft University of Technology, Delft, The Netherlands. [3]Netherlands Organization for Applied Scientific Research TNO, The Hague, The Netherlands. ✉e-mail: k.li@cml.leidenuniv.nl

impacts of treating imported plastic waste in the USA, using widely varying recycling rates from 8.7 to 50%[15]. Bishop et al. faced a lack of official data on exported plastics from Europe, leading them to use a broad range of recycling rates from 50 to 90%[16]. This reliance on domestic or scenario-based rates highlights the urgent need for comprehensive and transparent data to guide policy and research effectively.

Moreover, replacing the recycling rate of imported plastic waste with the domestic average is questionable for two main reasons. Firstly, domestic plastic waste often comes from diverse sources, resulting in heterogeneous and difficult-to-recycle mixtures, particularly in regions with inadequate or partial waste separation. In contrast, imported plastic waste is typically more concentrated and uniform, as it is pre-selected for exporting. Secondly, the UN Comtrade database shows that importing countries pay for plastic waste, indicating its economic value (see Fig. 1)[17]. If these imports were not processed into valuable recyclates−i.e., if they were primarily dumped or burned−the importing companies would face significant financial losses, making it unsustainable for them. Therefore, any viable approach must ensure that at least part of the imported plastic is converted into economically valuable outputs through recycling to offset initial costs.

In this work, we introduce a novel approach by defining the *Required Recycling Rate* (*RRR*). We estimate the *RRR* for the 22 largest plastic waste-importing countries from 2013 to 2022 based on the economic break-even point, where the revenue from recycling matches the costs of imports and the recycling process (labour, electricity, and real estate rentals[18,19]). We assume that recyclates can be sold at prices comparable to primary plastics and consider physical losses throughout the recycling process[20,21]. Import costs and primary plastic values are derived from 186,861 bilateral trade records for four plastic wastes (PE, PS, PVC and others) and six primary plastics (HDPE, LDPE,

PS, PVC, PET and PP) from the UN Comtrade database. Here, 'recycling' specifically refers to mechanical recycling, the predominant method for recycling imported waste in Global South countries[22,23]. Our findings indicate that the *RRR* for imported plastic waste in the 22 research countries significantly surpasses their reported national recycling rates. Sensitivity and Monte Carlo-based uncertainty analyses further confirm the robustness of these results.

## Results

### Required recycling rates across four plastic wastes and 22 countries

Our analysis shows that at least 63% of the imported plastic waste must be recycled to offset the costs. However, the *RRR*s vary across countries and plastic waste types (Fig. 2).

Due to the significant gap between recycling costs and product prices, countries in Asia and Eastern Europe have the lowest *RRR* for imported plastic waste, starting at around 40%. Specifically, Thailand, Turkey, and the Czech Republic have the lowest *RRR* benchmarks for their respective plastic waste types, ranging from 40 to 50%. In contrast, higher *RRR*s are needed for Western Europe and North America, reflecting limited profitability for recycling imported plastic waste in these regions. The highest *RRR*s are observed in France, the UK, Belgium, and Canada, with average values between 61% and 82% for all plastic waste types. For comparison, the mechanical recycling costs collected from other literature sources are detailed in Supplementary Table 2.

Examining distributions among four plastic waste types reveals that PE and PS waste have the lowest *RRR*, averaging 10–20% lower than those for PVC and 'Others' within the same region. PVC recycling is already hindered by challenges such as its chlorine content and contamination issues[24]. Our results further imply that recycling PVC is

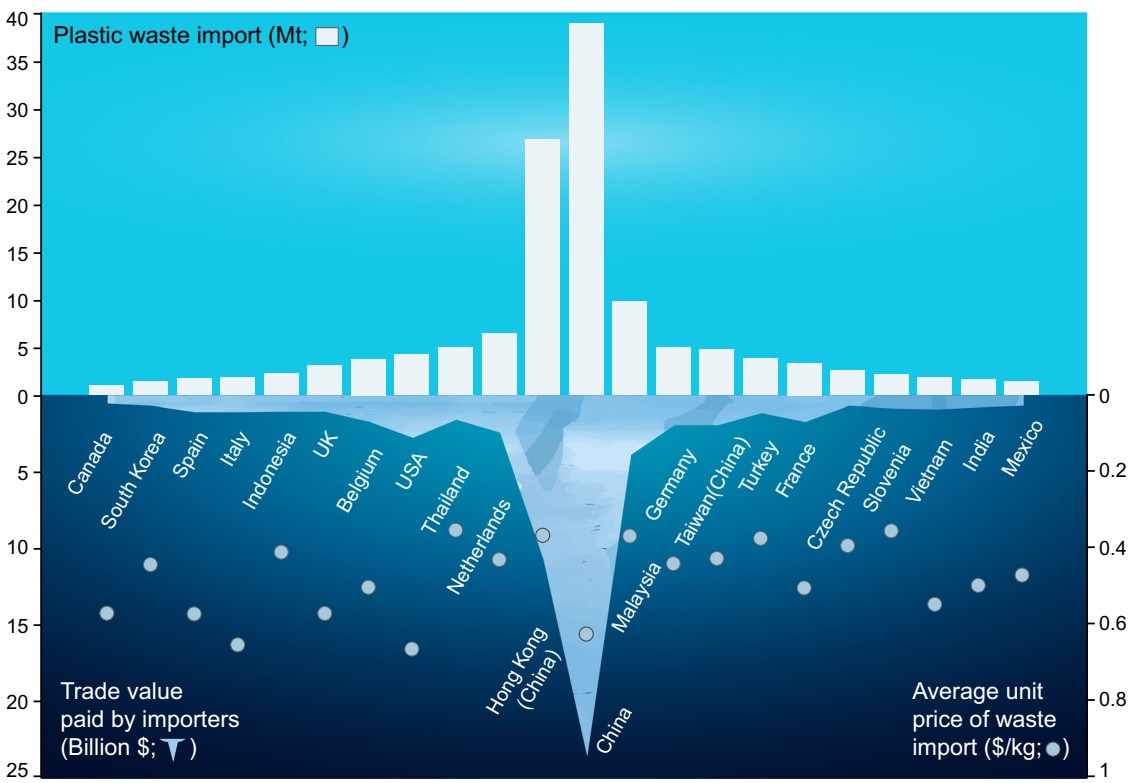

**Fig. 1 | Total plastic waste imports, total trade values, and average unit prices paid by the top 22 importers from 2013 to 2022.** The trade value reflects the cost, insurance and freight (CIF) price. The average unit price of waste import is calculated based on the weighted values of four plastic waste types (PE, PS, PVC and others) across years. The original trade data, including net weight and trade value, are sourced from the UN Comtrade database. The volume of plastic waste import is reported in net weight (million tonnes, Mt).

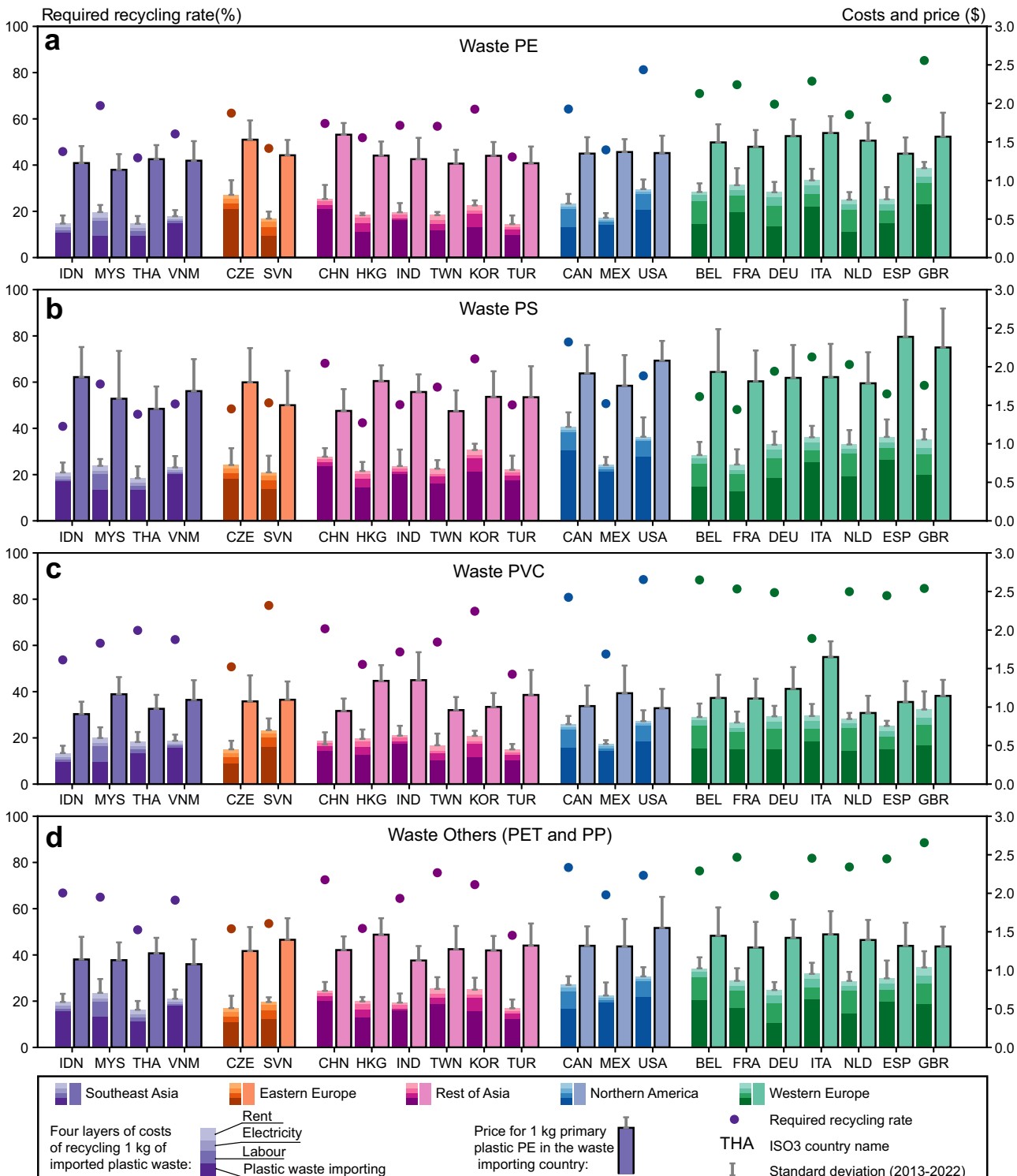

**Fig. 2 | Required recycling rate of imported plastic waste by country and plastic waste type.** The *RRR* is displayed for waste PE (**a**), waste PS (**b**), waste PVC (**c**), and waste 'Others' (**d**). The 22 research countries are geographically divided into five country groups. For each country, the left bar represents the costs associated with recycling 1 kg of plastic (including plastic waste imports), while the right bar shows the value of 1 kg of recycled plastic. The *RRR* calculated using mirror trade data is shown in Supplementary Fig. 1, with comparisons detailed in Supplementary Table 1. The annual *RRR* from 2013 to 2022 across 22 research countries and four plastic waste types is presented in Supplementary Figs. 2–5.

less economically competitive compared to other plastics, due to a narrower profit margin between recycling costs and primary plastic prices. This results in *RRR*s for PVC that are on average 14% higher than for PE from 2013 to 2022. The higher *RRR*s for 'Others' plastic waste (Fig. 2d) are attributed to greater variability in recycling costs

and primary plastic prices, with mixed plastic waste falling into this category.

The variation in *RRR* across different types of plastic waste serves as a crucial market signal for each country's plastic waste import structure. For example, the Netherlands demonstrates a significant

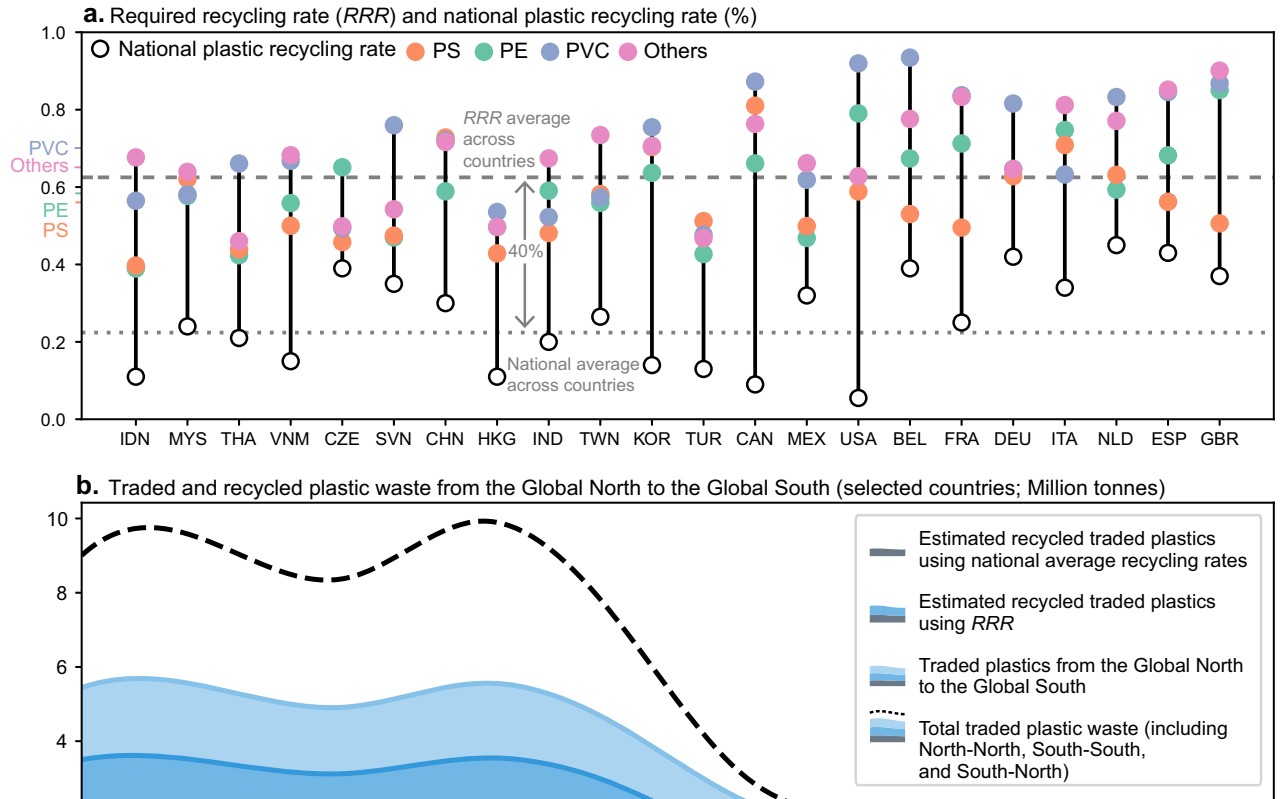

**a.** Required recycling rate (*RRR*) and national plastic recycling rate (%)

**b.** Traded and recycled plastic waste from the Global North to the Global South (selected countries; Million tonnes)

**Fig. 3 | The difference between required recycling rates and national plastic recycling rates across 22 countries. a** Illustrates the variations in average *RRR* across countries and plastic types. The dashed line represents the average *RRR* across countries, weighted by the total import mass across countries. The plastic waste type label on the *Y* axis displays the average *RRR* of each plastic waste type, weighted by the import mass of each plastic waste type across countries. In addition, the dotted line below denotes the country's average plastic recycling rate, weighted by the annual domestically generated plastic waste across countries.

Mass data corresponding to (**a**) are provided in Supplementary Table 3. A comparison of the average *RRR*, weighted by either trade mass or trade value and using both trade and mirror trade data, is shown in Supplementary Table 5. **b** Illustrates how discrepancies between these two recycling rates affect the estimates of recycled plastics from the waste traded from the Global North to the Global South between 2013 and 2022 (countries involved are listed in Supplementary Table 4). The trade data originate from the UN Comtrade database. The results from (**a**, **b**) calculated using mirror trade data are shown in Supplementary Fig. 6.

contrast in *RRR* between waste PVC and waste PE, with *RRR* of 83% and 62%, respectively. This difference suggests implicitly higher recycling costs and narrower profit margins in the PVC recycling market compared to the PE recycling market in the Netherlands. Confirming this trend, the Netherlands evidenced higher imports of waste PE (3 Mt) compared to waste PVC (0.1 Mt) during the period 2013–2022. Similar import structures are observed in countries such as Germany, the USA, France, and Belgium. In contrast, *RRR* differences across plastic waste types are less pronounced among countries in the Global South. For instance, *RRR*s across four plastic waste types range from 50–64% in Vietnam, 40–50% in Turkey, and 50–64% in India. Supplementary Table 1 provides a detailed comparison of *RRR* differences among plastic waste types across countries.

Although import costs are the largest component of overall expenses and are often seen as a major factor influencing plastic waste trade[25], they do not fully explain the observed differences in *RRR* between Europe and Asia as effectively as labour costs do. For instance, Germany, a major plastic waste importer in Western Europe, faces import costs ranging from $0.33 to $0.57 per kilogram across the four plastic types assessed. These costs are only slightly higher than those of large Asian importers such as Turkey ($0.27–$0.53) and Thailand ($0.27–$0.46). In contrast, labour costs, the second largest cost factor, are significantly higher in Western Europe. Recycling 1 kg of imported

plastic waste in Germany incurs average labour costs of $0.26 from 2013 to 2022, which is approximately four times higher than in Turkey ($0.067) and five times higher than in Thailand ($0.052). It is important to note that these cost-related statistics, collected at the country level, may not fully capture regional variations within countries.

## Comparison between the required recycling rate and the domestic average

A notable discrepancy emerges when comparing the calculated *RRR* with the collected national plastic recycling rates across 22 countries (see Supplementary Table 3). The *RRR* averages 63% for the period between 2013 and 2022, which is 40% higher than the average national plastic recycling rate of 23% (Fig. 3a). The *RRR* average and the domestic average are weighted by import mass across countries and plastic waste types and by the annual domestic plastic waste generated across countries, respectively (refer to Supplementary Table 3). This discrepancy subsequently affects the estimation of the amount of plastics recycled from the waste traded from the Global North to the Global South (refer to Fig. 3b, with the countries involved detailed in Supplementary Table 4). Using the national plastic recycling rate, the annual amount of recycled plastics from the plastic waste trade from the Global North to the Global South averaged 0.37 million tonnes per year (Mt yr⁻¹) over the past five years (2018–2022). In contrast, the

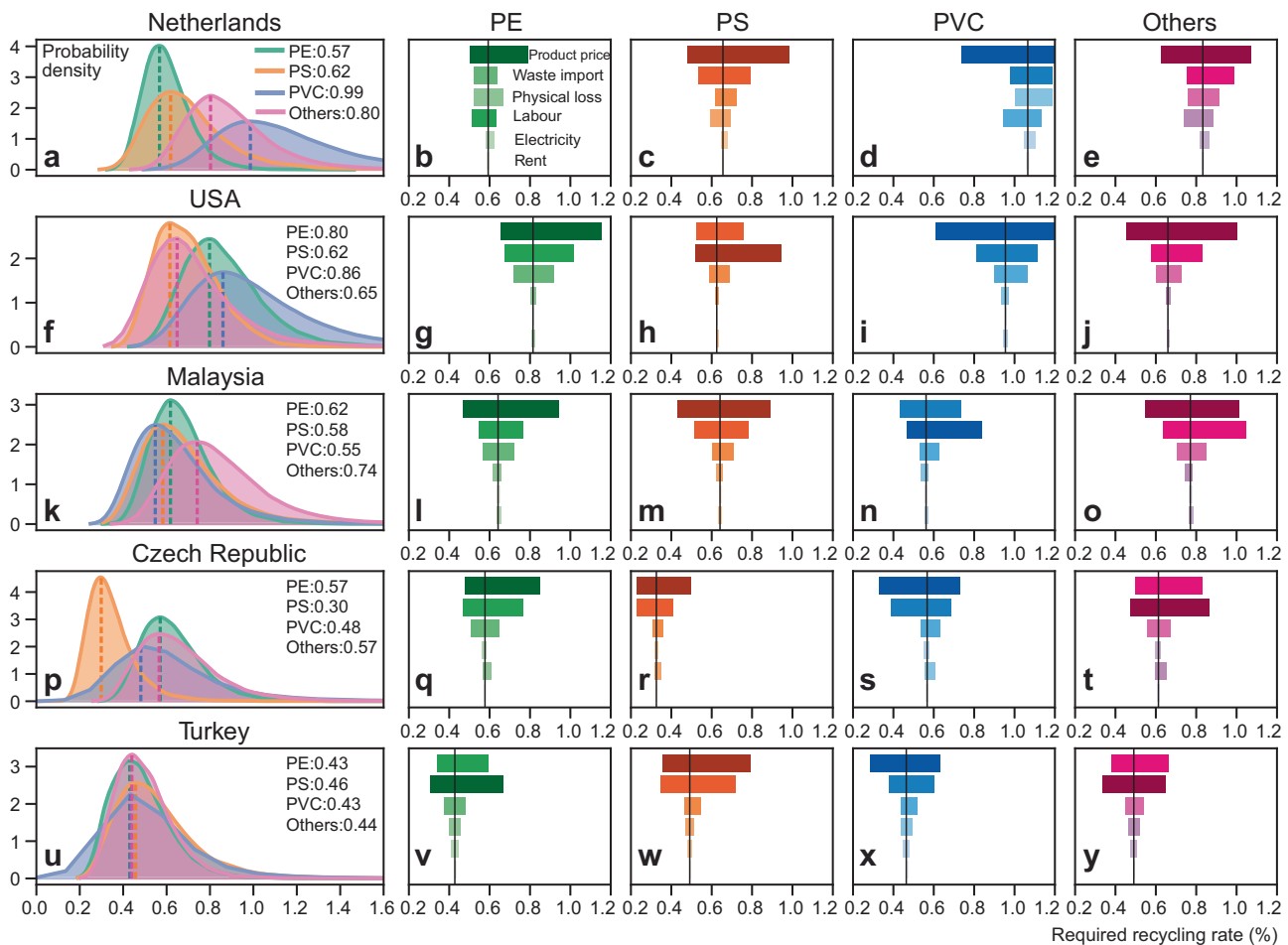

**Fig. 4 | Sensitivity analysis and Monte Carlo simulation of required recycling rate.** The selected countries include the Netherlands (**a**–**e**), the USA (**f**–**j**), Malaysia (**k**–**o**), the Czech Republic (**p**–**t**), and Turkey (**u**–**y**). Sensitivity analysis and Monte Carlo simulation results for other countries are presented in Supplementary Figs. 7–15. The length and colour depth of the horizontal bars represent the range of sensitivity results. The variable 'product price' refers to the value of recycled plastics.

annual recycling volume surges to 1.04 Mt yr⁻¹ if the *RRR* is used, an increase of 0.67 Mt, roughly equivalent to France's recycled plastics output in 2022[26].

**Assessing uncertainty in the required recycling rate**

We conducted a sensitivity analysis to examine how six key variables influence our results by country and plastic waste type (Fig. 4). The analysis considers both pessimistic and optimistic scenarios, representing each variable based on their minimum and maximum values observed from 2013 to 2022.

On average, fluctuations in the product price for recycled plastics have the greatest impact on the calculated *RRR*, varying between −25% and +36%. Variations in *RRR* are also significantly affected by import costs, with changes ranging from −21% to +29%, and physical losses, ranging from −8% to +11%. Labour costs contribute to fluctuations of −4% to +4%, electricity costs from −1% to +2%, and rental costs from −0.9% to +1%. Regional differences are particularly notable in labour costs, with pronounced variations between Europe and Asia. For example, the Netherlands shows fluctuations ranging from −11% to +6% (Fig. 4b–e), compared to narrower ranges of −4% to +2% in Malaysia and −1% to +1% in Indonesia. In addition, the analysis reveals notable variations across waste types, particularly for PVC and 'Other' types. The variations in *RRR* of these two waste types are also attributed to fluctuations in product prices and import costs.

## Discussion

The divergent recycling rates across countries result in varied estimations of recycled volumes from the plastic waste trade, complicating the assessment of its environmental impacts. Notably, the *RRR* averaged ~63% across 22 major importers and four plastic waste types from 2013 to 2022, significantly higher than the average domestic recycling rate of 23%. Moreover, country-specific *RRR* values exceed those reported in previous studies based on domestic recycling rates. For example, while Wen et al.[8] assumed a recycling rate of 38% for imported plastic waste in Malaysia for 2018, our study indicates a minimum required recycling rate in Malaysia of 58% (PE and PVC), 62% (PS), and 64% (Others) over the period from 2013 to 2022. Such variations in recycling rates can lead to differing estimates of the environmental impacts associated with the plastic waste trade. Higher recycling rates suggest reduced emissions from the avoided virgin plastic production[27]. For instance, Wen et al.[8] assessed the environmental impact of China's plastic import ban using domestic recycling rates for imported plastics, estimating the net carbon emissions of treating traded plastic waste in 2018 at 0.13 Mt CO₂-eq. In a scenario reflecting a 50% increase in countries' recycling rates, closely aligning with our calculated *RRR* for the same period, this figure dropped to −60 Kt CO₂-eq.

*RRR* enhances the accuracy of modelling the fate and impacts of traded plastic waste, which is crucial for scientific research and policy implementation. By indicating the proportion of recycling versus non-

recycling, *RRR* provides valuable data for assessing the environmental impacts of the global plastic waste trade, particularly in waste-importing countries. Moreover, the annual *RRR* data across countries and plastic waste types sheds light on how external events influence the global plastic waste trade. For instance, a notable increase in *RRR* across many countries in 2020 coincided with a drop in crude oil prices[28], suggesting that lower prices for virgin and recycled plastics necessitated a higher *RRR* to cover costs and achieve profitability. In terms of policy implications, *RRR* can assist waste-importing countries in formulating and adjusting their recycling targets. Instead of relying on domestic recycling rates, which are often based solely on domestically generated plastic waste, countries should consider separate targets for imported plastic waste, recognising their distinct characteristics.

Our research indicates that while the average *RRR* of 63% is higher than the domestic average of 23% across 22 research countries, it still falls short of ideal recycling rates. This gap suggests a significant portion of traded plastics may be mismanaged[29]. To address this, transparent tracking systems, such as a robust prior informed consent procedure[30], are essential. The OECD control system for waste recovery serves as a notable example, requiring disclosure of pre-consented recovery facilities and technologies in waste-importing countries[31]. Although recycling costs may be higher in developed countries, the overall environmental impact is often lower compared to that in Southeast Asia. These environmental concerns are reflected in the EU's newly adopted waste shipment regulation, which bans plastic waste exports to non-OECD countries starting in November 2026[13].

Our approach to calculating the *RRR* focuses on primary cost factors, providing a minimum benchmark for recycling imported plastic waste. The actual *RRR* might be higher when considering additional costs like environmental costs, capital investment, and operational expenses (e.g., chemical feedstocks[18], maintenance[32] and value-added taxes[19]). Although limited cost factors may underestimate the *RRR*, this method aligns with our goal of establishing a minimum benchmark, providing a better calibration than the domestic recycling rates previously used for imported plastic waste. Future research should explore the full costs and benefits of imported plastic waste for a more comprehensive *RRR* assessment.

Due to data constraints, we used primary plastic exports as a proxy for recycled plastic revenue to ensure consistency. However, advancements in recycling technologies (e.g., chemical, enzymatic and solvent-based methods) may create higher-value products not captured by current primary plastic classifications[33,34], potentially leading to an overestimation of the *RRR* in some developed countries. In addition, the four HS codes under 3915 may not fully reflect the quality and diversity of plastic waste, indicating a need for expanded classification coverage.

While our work provides valuable insights into country-specific recycling rates, it does not address regional disparities within countries. Variations in costs such as electricity, labour, and rent can be significant within a country, underscoring the need for future research to determine *RRR* at regional and city levels for more localised policymaking. Caution is also advised when applying the *RRR* to estimate recycled volumes and environmental impacts in trade transit countries or regions like Hong Kong (China), as inaccurate trade data may lead to errors in the *RRR* calculation. Prioritising actual recycling rates through mass balance is recommended for precision, though the *RRR* remains useful for addressing data gaps and estimating rates where physical measurement is impractical.

## Methods

### Accessing the required recycling rate

Importers aim to profit from recycling plastic waste, but face uncertainty since the recyclability of the waste is often unknown until the container is opened[35]. They must balance maintaining sufficient plastic waste feedstocks to ensure consistent recycling production while keeping both importing and recycling costs below the market value of secondary plastics. The cost factors include expenses related to plasitc waste import, labour, electricity, rent, and physical losses during recycling. We link the recycling rate of imported plastic waste to a cost-benefit inequality (Eqs. (1)–(3)) spanning 2013 to 2022, where costs should be less than or equal to benefits. For each year within this period, we selected importing countries that accounted for 70% of the global plastic waste imports, resulting in a total of 22 research countries. This equation can be enhanced with regional data on costs, providing a more accurate reflection of *RRR* across geographical units.

$$\left(\sum_p W_{i,p,c,t} \times PI_{i,p,c,t}\right) + C_{i,c,t} \leq \left(\sum_p W_{i,p,c,t}\right) \times (1 - Q_i) \times R_{i,c,t} \times PR_{i,c,t}$$

(1)

$$C_{i,c,t} = LAB_{c,t} + ELE_{i,c,t} + RET_{c,t}$$

(2)

$$R_{i,c,t} \geq \frac{PI_{i,c,t} + c_{i,c,t}}{(1 - Q_i) \times PR_{i,c,t}}$$

(3)

Where $W_{i,p,c,t}$ indicates the net weight of the imported plastic waste of type $i$ (referring to one of four waste plastics documented in the harmonised system (HS): PE, PS, PVC and others) from the country $p$ to the country $c$ of the year $t$; $PI_{i,c,t}$ indicates the per-unit price of the imported plastic waste of type $i$ by country $c$ in the year $t$; The upper-case $C_{i,c,t}$ denotes the operational costs during the mechanical recycling of plastic waste $i$ in the importing country $c$ of the year $t$, including costs for labour ($LAB_{c,t}$), electricity ($ELE_{i,c,t}$), and rent ($RET_{c,t}$) in Eq. (2). $Q_i$ indicates the physical loss of plastic waste of type $i$ during mechanical recycling. $R_{i,c,t}$ indicates the recycling rate of imported plastic waste of type $i$ in the country $c$ of the year $t$; $PR_{i,c,t}$ indicates the per-unit price of recycled plastic of type $i$ in the importing country $c$ of the year $t$. The lower-case $c_{i,c,t}$ denotes the per-unit operational cost when dividing the $C_{i,c,t}$ by $\sum_p W_{i,p,c,t}$.

The minimum recycling rate of imported plastic waste, enabling an economic break-even point, is referred to as the *Required Recycling Rate* (*RRR*). To derive a consistent unit price of recycled plastics, we used the trade data for plastics in primary forms (i.e. plastic pellets, flakes, etc.) recorded in the UN Comtrade database from 2013 to 2022[36], which consists of both virgin and secondary plastics. Within the same database, we also accessed the trade data for plastic waste. Both unit prices for plastic waste and primary plastics are determined by dividing the trade values and the net weights between trading countries. Moreover, given that primary plastic includes a broader range of polymer subcategories than the four plastic wastes (waste PE, PS, PVC and others), we further map waste PE with the primary plastics HDPE and LDPE, using a share factor that varies by country (Supplementary Table 6). Similarly, the 'others' waste category is mapped to primary plastics PET and PP. The unit price of imported plastic waste and primary plastic are calculated as follows:

$$PI_{i,c,t} = \frac{\sum_p V_{i,p,c,t}}{\sum_p W_{i,p,c,t}}$$

(4)

$$PR_{i,c,t} = \begin{cases} \frac{\sum_p V_{PS,p,c,t}}{\sum_p W_{PS,p,c,t}} \left(or \frac{\sum_p V_{PVC,p,c,t}}{\sum_p W_{PVC,p,c,t}}\right) & \text{if } i \text{ refers to waste PS } (or \text{ waste PVC}) \\ \frac{\sum_p V_{LDPE,p,c,t}}{\sum_p W_{LDPE,p,c,t}} \times r_{LDPE,c} + \frac{\sum_p V_{HDPE,p,c,t}}{\sum_p W_{HDPE,p,c,t}} \times r_{HDPE,c} & \text{if } i \text{ refers to waste PE} \\ \frac{\sum_p V_{PET,p,c,t}}{\sum_p W_{PET,p,c,t}} \times r_{PET,c} + \frac{\sum_p V_{PP,p,c,t}}{\sum_p W_{PP,p,c,t}} \times r_{PP,c} & \text{if } i \text{ refers to waste others} \end{cases}$$

(5)

Where $V_{i,p,c,t}$ and $W_{i,p,c,t}$ indicate the trade value and net weight of imported plastic waste of type $i$ from the country $p$ to country $c$ in the year $t$; $V_{ps,p,c,t}$ and $W_{ps,p,c,t}$ (also subscripts for PVC, HDPE, LDPE, PET, and PP) indicate the trade value and net weight of six types of primary plastic exported from country $c$ to country $p$ in the year $t$, respectively. By grouping HDPE and LDPE as PE, and PET and PP as 'Others', with the share factors of $r_{HDPE,c}$, $r_{LDPE,c}$, $r_{PET,c}$, and $r_{PP,c}$ in the country $c$, six primary plastics are mapped to four plastic waste types.

## Processing bilateral trade data

We analysed 186,861 bilateral trade entries of plastic waste and primary plastics reported from both 22 research countries and their trading partners from 2013 to 2022 in the UN Comtrade database. These entries include trade value and net weight for four plastic waste types (waste PE in Harmonized System (HS) code 391510, waste PS in HS 391520, waste PVC in HS 391530, waste others in HS 391590) and six primary plastic types (HDPE in HS 390120, LDPE in HS 390110, Expandable PS in HS 390311, PVC in HS 390410, PET in HS 390760, PP in HS 390210).

Each trade entry typically includes details such as reporting country, partner country, period, net weight, and trade value. Ideally, each trade flow should be reported by both importer and exporter during the same period, with closely aligned net weight and trade values. However, discrepancies often arise due to varying reporting conventions; exporters typically report trade values as Free On Board (FOB), while importers report them on a Cost for Insurance and Freight (CIF) basis. For our analysis, we require trade values for a country's imported plastic waste and its exported primary plastics. There are two options: using trade values reported by the research country, including plastic waste import (CIF basis) and primary plastics export (FOB basis), or using mirror trade values reported by the trading partner of the research country, including plastic waste export (FOB basis) and primary plastics import (CIF basis). In calculating the *RRR* via the cost-benefit equation, we aim to incorporate the international transport cost for importing plastic waste while excluding the transport revenue for recycled primary plastics. Therefore, we prioritise using plastic waste imports (CIF basis) and primary plastics exports (FOB basis), both reported by the research countries. However, for a robustness check, we also include the *RRR* results based on mirror trade data reported by the trading partners of the 22 research countries in Supplementary Table 1.

We detected trade value outliers through a distributional analysis of the value-to-mass ratio for all trade entries by plastic type and year[37–39]. Since most of these unit price distributions follow a log-normal pattern, we transformed them into normal distributions by taking the natural logarithm (ln($/kg))[40]. By identifying outliers greater than three standard deviations from the mean value, -2.4% and 2.6% of plastic waste and primary plastic trade entries were flagged as outliers, respectively.

After organising the trade values by research country, year, and plastic type, any empty trade values were replaced with mirror data when available. For example, if the trade values of China's waste PVC imports in 2022 were missing, the corresponding export values by its trading partners for 2022 were used instead. This replacement accounted for 2.6% and 2.5% of grouped plastic waste and primary plastic entries, respectively. Further details regarding the stepwise changes in trade entries when processing the original trade data are provided in Supplementary Table 7.

## Costs and physical loss

The complete costs from importing plastic waste to producing recycled plastics include plastic waste imports, operational costs (electricity, labour, land rent, water, fuel, transportation, maintenance), fixed asset investments (buildings, machinery, equipment), potential environmental costs, and taxes[18,19,41]. Based on the work of Uekert

et al.[19], Larrain et al.[18] and Faraca et al.[41], we consider the four costs consistently across research countries from 2013 to 2022: imports (as indicated by the UN Comtrade database), electricity, labour and rent.

The labour cost for recycling 1 kg of plastic waste was calculated by multiplying the labour input intensity (the required person-hours to recycle one kilogram of plastic waste) by the hourly earnings of employees in each country. The labour input intensity was determined by the recycling company's annual output and its number of employees, sourced from voluntary disclosures on independent recycling company websites and reports by industry associations, as presented in Supplementary Table 8. The production rate (expressed in kg/person-hours) was derived by dividing the company's annual recycling output by the number of its employees and the yearly working hours, which are standardised as 8 h a day and 365 days a year. Subsequently, the labour input intensity was obtained by taking the inverse of the production rate, and these values are averaged at the country level (see Supplementary Table 8). The hourly earnings of employees (by manufacturing industry) during 2013–2022 across countries were referenced from the statistics on 'Average monthly earnings of employees by sex and economic activity (annual)' by the International Labour Organization[42], as shown in Supplementary Table 9.

The electricity cost for recycling per-unit plastic waste is derived by multiplying per-unit electricity consumption varied by plastic wastes and the industrial electricity price across countries and years. We sourced the electricity consumption per plastic waste in mechanical recycling from the life cycle inventories (Ecoinvent and LCA Commons) and other literature (detailed in Supplementary Table 10). For European countries, the industrial electricity tariffs were obtained from Eurostat (see Supplementary Table 11)[43]. For other countries, the tariffs were gathered from governmental documents or power company announcements, as shown in Supplementary Table 12.

The rent for recycling 1 kg of plastic waste is based on the area required to recycle 1 kg of plastic waste per year ($m^2$/kg*yr), which is calculated by dividing the land area occupied by a recycling company (plants are included) by its annual recycling output (see Supplementary Table 13). This value is subsequently multiplied by the annual industrial rent across countries and years, primarily sourced from either yearly or quarterly reports of real estate companies (see Supplementary Table 14). The physical losses of four plastic wastes during mechanical recycling stem from prior literature (see Supplementary Table 15), where the input of plastic waste and the output of recycled plastic are collected.

## Sensitivity and uncertainty analyses

A one-at-a-time sensitivity analysis was conducted to determine how the alteration of key variables impacts the *RRR*. The six key variables include four costs (imports, labour, electricity, and rent), product price, and physical loss during the mechanical recycling process. Two values (the minimum and maximum value) presenting the pessimistic and optimistic cases for each variable were selected in a 10-year series (2013–2022) across countries.

The probability range of the *RRR* was studied with a Monte Carlo simulation, where all six variables are included. Referring to Larrain et al.[18], the product price of recycled plastic is modelled by the normal distribution, with the value of mean and standard deviation evaluated from the 10-year time series across countries. The costs for labour, electricity, rent, and physical loss during the recycling process are assumed to have a Pert distribution, with the minimum value, maximum value, and the most likely value (median value) selected across countries. The import cost is considered to fit a modified Pert distribution with a most likely value weight equivalent to the minimum and maximum value[44]. The resulting uncertainties are propagated with a Monte Carlo simulation (sampling of 30,000) using kernel density smoothing[45].

**Reporting summary**

Further information on research design is available in the Nature Portfolio Reporting Summary linked to this article.

## Data availability

The required recycling rate data by country, year, and plastic waste type generated in this study have been deposited under Zenodo https://doi.org/10.5281/zenodo.8328894, along with other supporting data. In case of questions or requests, please contact K.L. Source data are provided with this paper.

## Code availability

All Python analysis codes and a catalogue have been deposited in Zenodo https://doi.org/10.5281/zenodo.8328894.

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

## Acknowledgements
K.L. would like to thank the support from the China Scholarship Council (No. 202006050026). Special appreciation is extended to Yanan Liang at Leiden University for her insightful discussions on the plastic waste trade.

## Author contributions
K.L., A.T., H.L., and H.W. designed the study. A.T. directed the research route. H.W. developed the calculation method and revised the manuscript. H.L. provided critical research ideas and examined the calculation method. K.L. collected the data, conducted the analysis and drafted the manuscript. All authors discussed the results and provided feedback on the manuscript.

## Competing interests
The authors declare no competing interests.
