## [Peer Review File · Nature Communications]

Economic viability requires higher recycling rates for imported plastic waste than expectedReviewers' Comments:

Reviewer #1:

Remarks to the Author:

Review of NCOMMS-23-60925

This manuscript applies an economic approach to deduce the “required recycling rate” for plastic waste importers, and shows that this rate is, on average, at least 64%. The manuscript is well written, with high quality graphics and descriptions and extensive supplementary information. I have just a few queries relating to methodology, and the authors are to be congratulated on their elegant approach to deduce an important data point in complex and somewhat opaque plastic waste management chains.

My main issue with the study is the central assertion that “Virtually all recently prominent published assessments assume that plastic waste from imports and domestic sources is treated identically”. This is used to imply that evidence to date on recycling rates for exported plastic is inaccurate - based on average domestic recycling rates of 23% vs the implied RRR of 64% found in this study. Yet, just four studies are cited to evidence the stated assumption. At least one prominent study I am aware of (<https://doi.org/10.1016/j.envint.2020.105893>), and I am sure others, took a value chain approach with explicit mid estimate recycling processor efficiencies of 50-90%, in line with the RRR reported here. Even at 64% recycling, circa 1 million tonnes of plastic waste imported to the global south ends up in local waste streams where it is prone to inadequate management. This is still a major problem for plastic accumulation in the environment (associated GHG emission implications cited in the Discussion are trivial in the scheme of things) - and consistent, rather than contrary to, numerous previous studies on this topic and recent policy developments. I therefore find the framing of the results a little misleading, and the novelty overstated.

Nonetheless, with more considered framing, the provision of a range of country-specific, empirically-derived RRR is valuable, and could be used widely for modelling plastic waste flows. I would suggest authors carefully reframe the manuscript to: (i) better represent previous plastic recycling studies, and avoid suggesting they primarily assume domestic recycling rates for imported plastic unless this can be very clearly demonstrated for a large number of studies; (ii) consider the alternative methods available to estimate recycling rates in import countries, such as application of mass balance data for recycling processors (e.g. Table S9); (iii) provide a more balanced summary of the impact and applicability of this work (e.g. to provide a new database of country-specific RRR that could be used in plastic waste studies).

Methodology questions

CIF needs elaboration in the Method section. It is unclear exactly what is included in the TRADECom import price data, and how compatible these data are with CIF.

It is stated that CIF data were missing in some cases – how extensive was this?

Correction for CIF was based on a ratio to FOB – what was this ratio, and how granular was it (by country, by plastic type, etc?)

The 0.5% and 0.7% stated relate to fraction of missing net weight data points?

Labour input intensities were estimated using data from company websites according to SI Table S2? How reliable and available are these data? The way this is written in the Method section makes it sound like just two cited techno-economic studies are used to derive a labour input intensity – please clarify.

Table S9 and the references therein provide an alternative way to estimate plastic losses during recycling based on mass flows – these input data to the analysis could in fact provide an alternative set of results that negate the need to back-calculate RRRs! That said, these references seem to mainly cover higher income countries, and the challenge is estimating practices in lower-income countries. But this is very important to discuss, in order to more comprehensively contextualise the significant and novelty of results in this manuscript.

Reviewer #2:

Remarks to the Author:

1. This paper uses trade data from the UN Comtrade database and techno-economic data from various sources to calculate the recycling costs of various types of plastics from an economic perspective, and creatively proposes the concept of required recycling rate (RRR) based on the traditional concept of recycling rate and the break-even point. The authors infer the minimum recycling rate of plastic recycling industry in various countries by RRR and came to an interesting conclusion that the previous claim that exporting plastic waste from developed countries to developing countries have a significant environmental impact may have been exaggerated. The paper is well written, the methodology is innovative and described in detail, and I expect the paper will be of interest to those in and out of the field. Further, I believe this paper can be accepted for publication, pending some major comments I provide below. These are concerned mostly with the interpretation of the results and conclusions.
2. I first advocate a brief review of previous research on plastic recycling rate. Recent publications underscore that less than 10% of plastic waste worldwide is recycled, and the continuous import of plastic waste from the Global North to the Global South has caused significant environmental impacts. Given this background, the conflation of domestic and imported plastic recycling rates appears manifestly unreasonable when analyzing a country's plastic recycling rate. A brief review of research related to plastic recycling rates can integrate existing methods and findings on recycling rates to demonstrate the shortcomings of existing research and emphasize the importance of this study.
3. The authors analyze the impact of factors such as labor, rent, and electricity on the RRR, and conclude that labor costs are a key factor in the differences of RRR between countries in Europe and Asia (Line 118-119). My main concern is that in the process of cost accounting, the author did not take into account the differences in labor levels and costs in different regions of the same country, which is particularly evident in some developing countries. They also conclude that the expected negative impact of plastic waste input from the Global North on the Global South may have been exaggerated. However, such an inference is only partial. The article only analyzes from an economic perspective, and further evaluation of its impact on the environment is needed to reach the above conclusion.
4. The article mentions that the RRR of different plastics varies greatly (Line 110-114). I think it is a

very important conclusion. Under the influence of economies of scale, the differences in the import structure of plastic waste in different countries will have a significant impact on their recycling costs. It is necessary to analyze the relationship between the required recovery rate and the import structure of plastic waste in different countries.

5. Lastly, what make me concern else is that the authors prefer the trade value reported by the research countries instead of comprehensively considering the mirror trade data of both sides, especially considering a series of studies offering diverse methods for processing trade data. I think the authors need to perform some processing on the raw data or show us some data quality by picture.

6. General comments:

1) Line 23-24: This statement is inappropriate. The article only discusses economic costs and required recycling rates, without evaluating the environmental impact of plastic waste trade.

2) Line 67-71: Different kinds of plastic waste in the Fig.1 can explain the import structure of different countries.

3) Line 109-117: The required recovery rate serves as a valuable indicator, enabling countries and governments to adjust their import structures promptly. This study appears to have overlooked the recycling structure of various countries. When a country imports a large amount of plastic waste with a high required recycling rate, it faces significant recycling pressure. I hope this aspect can be reflected in the article.

4) Line 150-155: The description of Fig.3(a) is a bit confusing, and I don't quite understand its meaning.

5) Line 179-183: The letters in the bottom left corner of the image are inconsistent in height, please unify the format. Please align the words in Fig.4 b

6) Line 186-187: I think that the adverse impacts of the plastic waste trade primarily stem from environmental costs rather than recovered costs. So why not consider evaluating the required recycling rate based on environmental data or both factors?

7) Line 239-251: Normally, for countries like China, there are significant differences in labor and rental costs between big and small cities. I hope this can be reflected in this article.

8) Line 293-301: Many previous studies have provided various methods for processing trade data. Like:

i. "Brewer, T. D. et al. A method for cleaning trade data for regional analysis: The Pacific Food Trade Database (version 2, 1995-2018). f6d6974d31f146110da3af7eee8f844f.pdf (windows.net)"

ii. "Chen. C. et al. Advancing UN Comtrade for Physical Trade Flow Analysis: Review of Data Quality Issues and Solutions. doi.org/10.1016/j.resconrec.2022.106526"

iii. "Szkutnik, T. et al. Identification of Outliers in High Density Areas with the Use of a Quantile Regression Model. CEEOL - Article Detail"

iv. Why does this study only refer to the data of the reporter country?

Reviewer #3:

Remarks to the Author:

The research perspective of the paper is unique. It tells an interesting fact that the results of the research have practical reference value for policy decision-making. The research method of the

paper is reasonable, the data is sufficient, the argumentation process is reasonable and the conclusion is scientific. The thesis logic is more rigorous, the style of writing is more rigorous. The following aspects need to be further improved in the paper: (1) the importance of the conclusion of the study is not fully demonstrated. (2) the value of the results is not reflected sufficiently (3) the analysis of the results is not thorough enough, and some important factors are neglected, such as the management cost of the circular supply chain, the cost of international transportation, the social overflow of plastic waste. (4) the scope of application of the conclusions needs to be given a clearer demonstration (5) the innovation point is presented more clearly (6) the discussion section adds some policy implications.

Point-to-point response to the reviewers' comments:

Reviewer #1 (Remarks to the Author):

This manuscript applies an economic approach to deduce the “required recycling rate” for plastic waste importers, and shows that this rate is, on average, at least 64%. The manuscript is well written, with high quality graphics and descriptions and extensive supplementary information. I have just a few queries relating to methodology, and the authors are to be congratulated on their elegant approach to deduce an important data point in complex and somewhat opaque plastic waste management chains.

We are glad to see that the reviewer appreciates the merits of our paper and the scale of this study.

My main issue with the study is the central assertion that “Virtually all recently prominent published assessments assume that plastic waste from imports and domestic sources is treated identically”. This is used to imply that evidence to date on recycling rates for exported plastic is inaccurate - based on average domestic recycling rates of 23% vs the implied RRR of 64% found in this study. Yet, just four studies are cited to evidence the stated assumption. At least one prominent study I am aware of (<https://doi.org/10.1016/j.envint.2020.105893>), and I am sure others, took a value chain approach with explicit mid estimate recycling processor efficiencies of 50-90%, in line with the RRR reported here.

We recognize that our previous emphasis on extended works relying solely on an average domestic recycling rate may have been inaccurate and overstated. Besides, we appreciate the valuable research provided by the reviewer that aligns with our findings, which we had previously overlooked. We have included a modest review of previous research in the introduction section, which is appended below your three suggestions.

Even at 64% recycling, circa 1 million tonnes of plastic waste imported to the global south ends up in local waste streams where it is prone to inadequate management. This is still a major problem for plastic accumulation in the environment (associated GHG emission

implications cited in the Discussion are trivial in the scheme of things) - and consistent, rather than contrary to, numerous previous studies on this topic and recent policy developments. I therefore find the framing of the results a little misleading, and the novelty overstated. Nonetheless, with more considered framing, the provision of a range of country-specific, empirically-derived RRR is valuable, and could be used widely for modelling plastic waste flows. I would suggest authors carefully reframe the manuscript to:

(i) better represent previous plastic recycling studies, and avoid suggesting they primarily assume domestic recycling rates for imported plastic unless this can be very clearly demonstrated for a large number of studies;

We have included a review of previous research in the introduction section and removed statements claiming extensive work has applied to the domestic recycling rate.

Rather than solely highlighting the gap in recycling rates between domestic and imported plastic waste, we emphasize the necessity for more transparent data to mitigate reliance on domestic or scenario-based recycling rates. The revised third paragraph in the introduction section (lines 48-61) is provided as follows:

“While the global plastic recycling rate remains low, there is an implicit assumption that traded plastic is primarily recycled¹³. However, determining the recycling rate for imported plastic waste in receiving countries is hindered by measurement difficulties, leading existing studies to rely on assumed domestic or scenario-based recycling rates, often lacking robust data support. For instance, Wen et al. quantified the shift in the environmental impact of plastic waste imports from China to Southeast Asia after 2017, assuming plastic recycling rates equalled domestic rates among five Southeast Asian countries, ranging from 10% to 40%⁷. Bourtsalas et al. faced similar challenges when estimating the environmental impact of treating imported plastic waste in the USA, assuming widely varying recycling rates ranging from the domestic recycling rate at 8.7% to the scenario-based 50%¹⁴. Bishop et al. encountered a lack of official data on the fate of exported plastics from Europe, prompting them to adopt a broader range of recycling rates from 50% to 90%¹⁵. This reliance on domestic or scenario-based recycling rates underscores the critical need for comprehensive, transparent data to inform policy and research efforts effectively”

(ii) consider the alternative methods available to estimate recycling rates in import countries, such as application of mass balance data for recycling processors (e.g. Table S9);

We appreciate the reviewer's attention to estimating recycling rates via mass balance. Please refer to our response addressing the final comment on methodology.

(iii) provide a more balanced summary of the impact and applicability of this work (e.g. to provide a new database of country-specific RRR that could be used in plastic waste studies).

We have treasured the three suggestions offered by the reviewer to better conclude and summarize our results. We have rephrased the contribution of this work by providing a dataset that could facilitate other research focusing on quantifying and mitigating the environmental impacts of plastic waste trade in both the abstract and introduction sections.

In the final sentence of the abstract (lines 21-23), we emphasized that “The country-specific RRR provided by this study could facilitate research and policy efforts aimed at quantifying and mitigating the environmental impact of plastic waste trade.”

In the final sentence of the review of previous work in the introduction section (lines 59-61), we indicate the research gap in the lack of transparent data: “This reliance on domestic or scenario-based recycling rates underscores the critical need for comprehensive, transparent data to inform policy and research efforts effectively.”

Methodology questions

CIF needs elaboration in the Method section. It is unclear exactly what is included in the TRADECom import price data, and how compatible these data are with CIF.

We would like to thank the reviewer for bringing up this lack of context. We are now describing the UN Comtrade database more extensively and explaining why we prefer CIF over FOB for primary plastics in calculating the required recycling rate.

This preference allowed us to calculate the *RRR* results using the trade data of the 22 research countries instead of their trading partners (mirror trade data). However, we have also included the *RRR* results calculated by using mirror trade data in Table S2. Finally, to enhance data clarity, we have included Table S3, which illustrates the stepwise trade entry changes during data processing, in the supplementary file.

Added in the methods section (lines 322-338):

“Each trade entry typically includes details such as reporting country, partner country, period, net weight, and trade value. Ideally, each trade flow should be reported by both importer and exporter during the same period, with closely aligned net weight and trade values. However, discrepancies often arise due to varying reporting conventions; exporters typically report trade values as Free On Board (FOB), while importers report them on a Cost for Insurance and Freight (CIF) basis.

For our analysis, we require trade values for a country's imported plastic waste and its exported primary plastics, derived from the UN Comtrade database. There are two options: using trade values reported by the research country, including plastic waste import (CIF basis) and primary plastics export (FOB basis), or using mirror trade values reported by the trading partner of the research country, including plastic waste export (FOB basis) and primary plastic import (CIF basis).

In calculating the *RRR* via the cost-benefit equation, we aim to incorporate the international transport cost for importing plastic waste while excluding the transport revenue for recycled primary plastics. Therefore, we prioritize using plastic waste imports (CIF basis) and primary plastics exports (FOB basis), both reported by the research countries. However, for a robustness check, we also include results based on mirror trade data reported by the trading partners of the 22 research countries in the supplementary file (Table S2).”

Furthermore, we have included the *RRR* results calculated by using mirror trade data (plastic waste exports and primary plastics imports reported by the trading partners of the 22 research countries) in the supplementary file for each figure in the results section. A comparison of the *RRR* results between these two trade data is now presented in Table S2.

Table S2. Comparison of the required recycling rate using trade data reported by 22 research countries and their trading partners (mirror trade data). The trade data from the 22 research countries include plastic waste imports and primary plastics exports, while the mirror trade data include plastic waste exports and primary plastics imports reported by the trading partners of the 22 research countries. The required recycling rate for each country was averaged over the period 2013-2022.

Country	PE		PS		PVC		Others	
	22 countries	Trading partners	22 countries	Trading partners	22 countries	Trading partners	22 countries	Trading partners
Belgium	0.709	0.696	0.537	0.630	0.883	0.820	0.763	0.748
Canada	0.642	0.680	0.774	0.777	0.808	0.841	0.778	0.763
China	0.580	0.563	0.681	0.571	0.672	0.640	0.725	0.606
Hongkong (China)	0.518	0.614	0.424	0.448	0.518	0.526	0.515	0.573
Czechia	0.625	0.599	0.484	0.502	0.507	0.566	0.513	0.482
France	0.748	0.842	0.482	0.532	0.844	0.851	0.822	0.812
Germany	0.663	0.665	0.647	0.655	0.829	0.852	0.658	0.704
India	0.572	0.532	0.503	0.439	0.572	0.627	0.644	0.583
Indonesia	0.459	0.447	0.409	0.468	0.537	0.509	0.668	0.549
Italy	0.763	0.786	0.709	0.685	0.630	0.696	0.818	0.835
Malaysia	0.657	0.593	0.592	0.491	0.609	0.584	0.650	0.656
Mexico	0.466	0.557	0.507	0.514	0.563	0.665	0.660	0.610
Netherlands	0.618	0.710	0.676	0.712	0.833	0.856	0.781	0.748
Taiwan (China)	0.568	0.546	0.578	0.472	0.614	0.584	0.756	0.631
Republic of Korea	0.642	0.572	0.701	0.612	0.748	0.680	0.704	0.725
Slovenia	0.472	0.474	0.510	0.386	0.773	0.651	0.536	0.542
Spain	0.689	0.690	0.549	0.505	0.815	0.847	0.816	0.780
Thailand	0.432	0.410	0.461	0.404	0.665	0.604	0.509	0.509
Turkey	0.435	0.397	0.502	0.446	0.475	0.406	0.485	0.447
USA	0.813	0.695	0.627	0.562	0.885	0.825	0.744	0.715
UK	0.852	0.800	0.586	0.655	0.846	0.890	0.885	0.894
Vietnam	0.535	0.480	0.506	0.503	0.625	0.460	0.637	0.528

To improve communication of data quality, we have added Table S3, which records all stepwise changes in processing the original trade entries from the UN Comtrade database.

Table S3. Stepwise changes during data processing in trade entries of plastic waste and primary plastics sourced from the UN Comtrade database.

Data processing	Plastic waste imports (22 research countries)	Plastic waste exports (trading partners of 22 research countries)	Primary plastics exports (22 research countries)	Primary plastics imports (trading partners of 22 research countries)
Trade value originally retrieved from the UN Comtrade database	21940 entries	20802 entries	70311 entries	73808 entries
Dropped Entries with empty net weight or trade value	21884 entries, 56 dropped (0.26% of the original trade)	20721 entries, 81 dropped (0.39% of the original trade)	70135 entries, 176 dropped (0.25% of the original trade)	73378 entries, 430 dropped (0.58% of the original trade)
Drop trade value outliers ¹	21366 entries, 518 dropped (2.36% of the original trade)	20273 entries, 448 dropped (2.15% of the original trade)	68335 entries, 1800 dropped (2.56% of the original trade)	71427 entries, 1951 dropped (2.64% of the original trade)
Grouped trade values by research countries, period, and plastic types	880 entries, 24 missing (2.73 % of the grouped entries)	880 entries, 1 missing (0.11% of the grouped entries)	1320 entries, 144 missing (10.91% of the grouped entries)	1320 entries, 33 missing (2.50% of the grouped entries)
Replacing empty trade value (if mirror data exists) ²	880 entries (23 replaced (2.62%), 1 missing (0.11%))	880 entries (0 replaced (0%), 1 missing (0.11%))	1320 entries (33 replaced (2.50%), 111 missing (8.41%))	1320 entries (0 replaced (0%), 33 missing (2.50%))

Note: ¹ Referring to Chatham House's method in identifying the bilateral trade outliers¹⁴, we assume that the logarithm of the unit price follows a normal distribution. Trade value outliers are identified by their calculated unit price greater than three standard deviations from the mean value. Please refer to the methodology section for further details.

It is stated that CIF data were missing in some cases – how extensive was this?

We have now provided a breakdown of the number of trade data points dropped or replaced at each step of the data processing in Table S3, as indicated above. After grouping trade values by research countries, period, and plastic types, we found that 2.73% and 10.91% of CIF-based trade values were missing for plastic waste imports and primary plastics exports, respectively. Subsequently, missing trade values were replaced with mirror trade values, if available, resulting in 0.11% and 8.41% of trade

values finally missing for plastic waste imports and primary plastics exports, respectively.

Correction for CIF was based on a ratio to FOB – what was this ratio, and how granular was it (by country, by plastic type, etc?)

We have now directly replaced the missing CIF-based trade values with available FOB-based trade values, as the replacement rate was identified to be only 2.62% and 2.50% for plastic waste imports and primary plastic exports, respectively.

We made all Python scripts related to data processing publicly available at the following link: <https://zenodo.org/records/10996867>

The 0.5% and 0.7% stated relate to fraction of missing net weight data points?

We have recalculated the missing number of net weights and trade value data, as indicated by the second row in Table S3. Currently, 0.26% and 0.25% of the original trade data, which are missing either net weights or trade values, have been dropped.

Labour input intensities were estimated using data from company websites according to SI Table S2? How reliable and available are these data?

We acknowledge the reviewer's concern regarding the use of company website data to estimate labour input intensity. The companies we included voluntarily disclose their employee numbers and recycling output on their websites, which are publicly accessible. Since all company website data are independently disclosed, cross-referencing is feasible for companies operating within the same country.

Additionally, we have augmented our dataset by incorporating data provided by industry associations across countries to facilitate cross-validation. These associations include the American Chemistry Council, the United States Environmental Protection Agency, the Recycling and Recoverers Association (Turkey), the China Plastic Recycling Association, and the PlastIndia Foundation (India).

Furthermore, we have extended the sample of related data. All this information is detailed in Table S4.

Table S4. Annual recycling output and number of employees in plastic recycling companies across countries. The data referring company's annual recycling output and number of employees are voluntarily disclosed on their independent websites, which are publicly accessible (accessed on April 24th, 2024). The person-hours to recycle 1 kg are subsequently averaged at the country level. Data for 'EU 27+3' is applied to nine European research countries plus South Korea and Hongkong (China). Canada shares the data of the United States.

Coverage	Data sources	References (or links)	Annual recycling output	Number of Employees	Production rate per person-hours (kg/person-hour)	Person-hours to recycle 1 kg
EU27+3	Industry Associations	Plastic Recyclers Europe (2022) ¹⁵	10.2 Mt	30000	116	0.0086
USA	Industry Associations	American Chemistry Council (2019) ¹⁶	6.5 Mt	24000	93	0.0108
USA	Government departments	United States Environmental Protection Agency (2020) ¹⁷	-	-	146	0.0070
USA	Company disclosure	Deltco Plastics (https://www.deltcoplastic.com/)	27.2kt	90	104	0.0096
USA	Company disclosure	Evergreen (https://www.evergreentogether.com/)	67kt	185	124	0.0081
USA	Company disclosure	Peninsula Plastics Recycling (https://peninsularecycling.com/)	27kt	92	101	0.0099
USA	Company disclosure	Atkore Northwest Polymers (https://nwpoly.com/)	18kt	60	103	0.0097
USA	Company disclosure	Resource Plastics Inc. (http://www.resource-plastics.com/)	34kt	100	117	0.0085
Mexico	Company disclosure	Indorama Ventures EcoMex, S de RL de CV (https://www.indoramaventures.com/en/worldwide/816/indorama-ventures-ecomex)	42kt	177	81	0.0120
Mexico	Company disclosure	ALPLA Recycling ¹⁸	15kt	70	73	0.0137
Turkey	Industry Associations	Recycling and Recoverers Association (GEKADER) (https://gekader.org.tr/)	12Mt	70000 ^a	59	0.0170
Turkey	Company disclosure	Folyopak Ambalaj San. Ve Tic. Inc.	18kt	140	44	0.0230

		(https://folyopak.com.tr/)				
Turkey	Company disclosure	Hür Plastik Geri Dönüşüm LTİ. STİ. (https://www.hurplastik.com/)	30kt	140	73	0.0140
Turkey	Company disclosure	Plasman Polimer San. Ltd. Şti. (https://www.plasman.com.tr/en/home-2/)	10kt	50	68	0.0150
China	Industry Associations	China Plastic Recycling Association (2022) ¹⁹	16Mt	100000	65	0.0150
China	Company disclosure	Zhejiang Haili Environmental Technology Co., Ltd. (https://www.hailirecycle.com/)	182kt	600	104	0.0096
India	Industry Associations	PlastIndia Foundation (2019) ²⁰	6Mt	100000 ^b	21	0.0476
India	Company disclosure	Ganesha Ecosphere Ltd. (https://ganeshaecosphere.com/)	119kt	1000	41	0.0240
India	Company disclosure	The Shakti Plastic Industries (https://www.shaktiplasticinds.com/)	120kt	500	82	0.0120
Malaysia	Company disclosure	Karich Sdn Bhd http://www.karich.com.my/plastic-recycling	15kt	270	19	0.0530
Malaysia	Company disclosure	Dragon Alliance Sdn. Bhd (https://www.daplastics.com/)	36kt	300	41	0.0244
Malaysia	Company disclosure	SG Green Resources Sdn. Bhd https://sggreen.com.my/index.html	6kt	70	29	0.0345
Taiwan (China)	Company disclosure	REMONDIS Taiwan (https://www.remondis-taiwan.com.tw/en/about-us/)	40kt	110	125	0.0080
Indonesia	Company disclosure	PT. Pradha Karya Perkasa (https://www.prakarsarecycling.com/)	30kt	320	32	0.0310
Indonesia	Company disclosure	Veolia Indonesia https://aqua.co.id/en/dan-one-aqua-and-veolia-indonesia-inaugurate-the-most-modern-and-largest-plastic-recycling-facility-in-indonesia	25kt	225	38	0.0260
Vietnam	Company disclosure	Công ty TNHH Công Nghiệp và Dịch Vụ Bình Minh (https://abmchemical.com/)	27kt	150	61	0.0160

Vietnam	Company disclosure	Vinatic Hai Phong Company Ltd. (https://www.vinatic.com.vn/pages/about-us)	60kt	250	82	0.0120
Thailand	Company disclosure	Billion Enterprise. Co., Ltd. (https://www.billionpolymer.com/)	10.8kt	75	49	0.020
Thailand	Company disclosure	Indorama Polyester Industries PCL ²¹	57kt	600 ^c	33	0.0303

Note: ^a With a total workforce of 350,000, it is assumed that the ratio of direct to indirect employment is 1:4¹⁶. ^b Together for direct and indirect employment. ^c Workers for polyester plastic production are included.

The way this is written in the Method section makes it sound like just two cited technological studies are used to derive a labour input intensity – please clarify.

We appreciate the reviewer's attention to this ambiguity. The two studies referenced in that section are intended to explain why we focused on labour, electricity, and rent as key operational costs in a plastic recycling company. They are not directly related to deriving the labour input intensity. We have rephrased the paragraph explaining how we calculate the labour input intensity in the main text. Please refer to lines 354 to 368 in the methods section:

“We consider three main operational costs during the mechanical recycling processes across research countries during 2013-2022, including labour, electricity and rent^{27, 29}. The labour cost for recycling 1 kg of plastic waste was calculated by multiplying the labour input intensity (the required person-hours to recycle one kilogram of plastic waste) by the hourly earnings of employees in each country. The labour input intensity was determined by the recycling company's annual output and its number of employees, sourced from voluntary disclosures on independent recycling company websites and reports by industry associations, as presented in Table S4. The production rate (expressed in kg/person-hours) was derived by dividing the company's annual recycling output by the number of its employees and the yearly working hours, which are standardized as 8 hours a day and 365 days a year. Subsequently, the labour input intensity was obtained by taking the inverse of the production rate, and these values are averaged at the country level (Table S4). The hourly earnings of employees

(by manufacturing industry) during 2013-2022 across countries were referenced from the statistics on ‘Average monthly earnings of employees by sex and economic activity (annual)’ by the International Labour Organization³⁶, as shown in Table S5.”

Table S9 and the references therein provide an alternative way to estimate plastic losses during recycling based on mass flows – these input data to the analysis could in fact provide an alternative set of results that negate the need to back-calculate RRRs! That said, these references seem to mainly cover higher income countries, and the challenge is estimating practises in lower-income countries. But this is very important to discuss, in order to more comprehensively contextualise the significant and novelty of results in this manuscript.

We thank the reviewer for emphasising the direct method to access recycling rates through mass balance. Indeed, it is a straightforward approach often considered when assessing the recycling rate of imported plastic waste. As the reviewer suggested, it is important to contextualize the contribution of our work by acknowledging the challenges posed by data constraints, particularly in developing countries where physical measurement data may be unavailable.

In response, we have provided additional context in the introduction section (lines 49 to 51), highlighting the rationale behind our work.

“While the global plastic recycling rate remains low, there is an implicit assumption that traded plastic is primarily recycled¹³. However, determining the recycling rate for imported plastic waste in receiving countries is hindered by measurement difficulties, leading existing studies to rely on assumed domestic or scenario-based recycling rates, often lacking robust data support.”

Moreover, in the discussion section (lines 258 to 262), we have underscored the preference for physical data obtained through mass balance when available.

“Finally, prioritizing the use of the actual recycling rate determined through mass balance is recommended due to its superior accuracy, whenever physical measurement is feasible. However, the RRR provides added value as it addresses data gaps and deduces recycling rates where physical measurement may not be practical.”

Reviewer #2 (Remarks to the Author):

1. This paper uses trade data from the UN Comtrade database and techno-economic data from various sources to calculate the recycling costs of various types of plastics from an economic perspective, and creatively proposes the concept of required recycling rate (RRR) based on the traditional concept of recycling rate and the break-even point. The authors infer the minimum recycling rate of plastic recycling industry in various countries by RRR and came to an interesting conclusion that the previous claim that exporting plastic waste from developed countries to developing countries have a significant environmental impact may have been exaggerated. The paper is well written, the methodology is innovative and described in detail, and I expect the paper will be of interest to those in and out of the field. Further, I believe this paper can be accepted for publication, pending some major comments I provide below. These are concerned mostly with the interpretation of the results and conclusions.

We are glad that the reviewer recognizes the strengths and significance of our paper. We are committed to making further improvements to enhance its quality.

2. I first advocate a brief review of previous research on plastic recycling rate. Recent publications underscore that less than 10% of plastic waste worldwide is recycled, and the continuous import of plastic waste from the Global North to the Global South has caused significant environmental impacts. Given this background, the conflation of domestic and imported plastic recycling rates appears manifestly unreasonable when analyzing a country's plastic recycling rate. A brief review of research related to plastic recycling rates can integrate existing methods and findings on recycling rates to demonstrate the shortcomings of existing research and emphasize the importance of this study.

We are grateful to the reviewer for identifying the inconsistent transition from the second to the third paragraph in the introduction section. We have incorporated a review of previous research to address this inconsistency. Rather than solely highlighting the gap in recycling rates between domestic and imported plastic waste, we emphasize the necessity for more transparent data to mitigate reliance on domestic or scenario-based recycling rates. The revised third paragraph in the introduction section is provided as follows (lines 48 to 61):

“While the global plastic recycling rate remains low, there is an implicit assumption that traded plastic is primarily recycled¹³. However, determining the recycling rate for imported plastic waste in receiving countries is hindered by measurement difficulties, leading existing studies to rely on assumed domestic or scenario-based recycling rates, often lacking robust data support. For instance, Wen et al. quantified the shift in the environmental impact of plastic waste imports from China to Southeast Asia after 2017, assuming plastic recycling rates equalled domestic rates among five Southeast Asian countries, ranging from 10% to 40%⁷. Bourtsalas et al. faced similar challenges when estimating the environmental impact of treating imported plastic waste in the USA, assuming widely varying recycling rates ranging from the domestic recycling rate at 8.7% to the scenario-based 50%¹⁴. Bishop et al. encountered a lack of official data on the fate of exported plastics from Europe, prompting them to adopt a broader range of recycling rates from 50% to 90%¹⁵. This reliance on domestic or scenario-based recycling rates underscores the critical need for comprehensive, transparent data to inform policy and research efforts effectively.”

3. The authors analyze the impact of factors such as labor, rent, and electricity on the RRR, and conclude that labor costs are a key factor in the differences of RRR between countries in Europe and Asia (Line 118-119). My main concern is that in the process of cost accounting, the author did not take into account the differences in labor levels and costs in different regions of the same country, which is particularly evident in some developing countries.

We appreciate the reviewer’s attention concerning the current approach cannot adequately reflect regional disparities in labour and rental costs. Please refer to our response to the reviewer’s general comment (7) for further details.

They also conclude that the expected negative impact of plastic waste input from the Global North on the Global South may have been exaggerated. However, such an inference is only partial. The article only analyzes from an economic perspective, and further evaluation of its impact on the environment is needed to reach the above conclusion.

We agree with the reviewer that we should carefully reframe our conclusions related to changes in environmental impacts. We acknowledge that we cannot forecast the

environmental consequences as we did not conduct a specific analysis for this purpose. We have adjusted the way to illustrate the relationship between the *RRR* results and the environmental impacts.

In the last sentence of the abstract section (lines 21-23), we have removed the forecast about environmental impact and emphasized our contribution in providing country-specific *RRR* that could facilitate research and policy efforts measuring and mitigating the associated environmental impacts:

“The country-specific required recycling rates provided by this study could facilitate research and policy efforts aimed at quantifying and mitigating the environmental impact of plastic waste trade.”

In the discussion section (lines 208-210), we have revised the sentence to focus on the relationship between recycling rates, recycled volume, and the measurement of environmental impacts associated with plastic waste trade:

“The divergent recycling rates across countries result in varied estimations of recycled volume within traded plastic waste, thereby complicating the measurement of environmental impacts linked to the global plastic waste trade.”

Additionally, in lines 216-217, we have removed the inaccurate description about directly linking recycling rates with the environmental impacts of plastic waste trade. Instead, we have rephrased the sentence to indicate that “Such variations in recycling rates can lead to fluctuation in the estimation of environmental impacts associated with plastic waste trade.”

4. The article mentions that the RRR of different plastics varies greatly (Line 110-114). I think it is a very important conclusion. Under the influence of economies of scale, the differences in the import structure of plastic waste in different countries will have a significant impact on their recycling costs. It is necessary to analyze the relationship between the required recovery rate and the import structure of plastic waste in different countries.

We appreciate the reviewer's insights into the *RRR* difference among four plastic waste types within a country. Please refer to our response to the reviewer's general comment (3) for further details.

5. Lastly, what make me concern else is that the authors prefer the trade value reported by the research countries instead of comprehensively considering the mirror trade data of both sides, especially considering a series of studies offering diverse methods for processing trade data. I think the authors need to perform some processing on the raw data or show us some data quality by picture.

Thanks for the reviewer's valuable suggestion on reflecting the *RRR* results using the trade data reported by the trading partners (mirror trade data). Please refer to our response to the reviewer's general comment (8) for further details.

6. General comments:

1) Line 23-24: This statement is inappropriate. The article only discusses economic costs and required recycling rates, without evaluating the environmental impact of plastic waste trade.

We have revised the original statement to offer a more balanced summary of the impact and applicability of this work. This sentence was revised as follows in the abstract section (lines 21-23):

“The country-specific required recycling rates provided by this study could facilitate research and policy efforts aimed at quantifying and mitigating the environmental impact of plastic waste trade.”

2) Line 67-71: Different kinds of plastic waste in the Fig.1 can explain the import structure of different countries.

We greatly appreciate the reviewer's insightful comments regarding the disparity in *RRR* among plastic waste types and its reflection on a country's plastic import structure. Inspired by this, we have added a new paragraph to the main text. Please refer to our reply under the general comment (3).

3) Line 109-117: *The required recovery rate serves as a valuable indicator, enabling countries and governments to adjust their import structures promptly. This study appears to have overlooked the recycling structure of various countries. When a country imports a large amount of plastic waste with a high required recycling rate, it faces significant recycling pressure. I hope this aspect can be reflected in the article.*

Thank you for providing us with such valuable insights regarding the variation in *RRR* among four plastic waste types at the country level. We agree that the disparity of *RRR* among plastic waste types could reflect varied plastic waste import structures within a country, as evidenced by the country's imports for specific waste types. We have incorporated a new paragraph into the results section (lines 124 to 136) to address this important aspect, as shown below:

“The variation in *RRR* across different types of plastic waste serves as a crucial market signal for each country's plastic waste import structure. For example, the Netherlands demonstrates a significant contrast in *RRR* between waste PVC and waste PE, with *RRRs* of 83% and 62% respectively. This difference suggests implicitly higher recycling costs and narrower profit margins in the PVC recycling market compared to the PE recycling market in the Netherlands. Confirming this trend, the Netherlands evidenced higher imports of waste PVC (3 Mt) compared to waste PE (0.1 Mt) during the period 2013-2022. Similar import structures are observed in countries such as Germany, the USA, France, and Belgium. In contrast, *RRR* differences across plastic waste types are less pronounced among countries in the global South. For instance, *RRRs* across four plastic waste types range from 50%-64% in Vietnam, 40%-50% in Turkey, and 50%-64% in India. Table S3 provides a detailed comparison of *RRR* differences among plastic waste types across countries.”

4) Line 150-155: *The description of Fig.3(a) is a bit confusing, and I don't quite understand its meaning.*

In the revised description, we have rephrased the sentence by starting with the graphic objects, including a dashed line above (the average *RRR* across countries), a dotted line below (the average domestic plastic recycling rate), and the plastic waste type labels on the y-axis (the average *RRR* of specific plastic waste type).

All three of these average values are weighted averages, calculated based on specific criteria. The average *RRR* across countries was weighted by the country's import mass during 2013-2022. The dotted line reflects the average domestic plastic recycling rate, which was weighted by the annual domestic plastic waste across countries. Lastly, the plastic waste type labels on the y-axis indicate the average *RRR* across different plastic waste types, weighted by the country's import mass of each plastic waste type during 2013-2022.

The revised description for Fig.3a is provided as follows (lines 171 to 177):

“Fig. 3(a) illustrates the variations in average *RRR* across countries and plastic types. The dashed line above represents the average *RRR* across countries, weighted by total import mass across countries. The plastic waste type label on the Y-axis displays the average *RRR* of each plastic waste type, weighted by the import mass of each plastic waste type across countries. Additionally, the dotted line below denotes the average recycling rate of domestically generated plastic waste, weighted by the annual domestic plastic waste across countries. Mass data corresponding to Fig. 3(a) are provided in Table S12.”

5) Line 179-183: The letters in the bottom left corner of the image are inconsistent in height, please unify the format. Please align the words in Fig.4 b

Thank you for pointing out the inconsistency. We have aligned the words in Fig.4 as you suggested.

6) Line 186-187: I think that the adverse impacts of the plastic waste trade primarily stem from environmental costs rather than recovered costs. So why not consider evaluating the required recycling rate based on environmental data or both factors?

We appreciate the reviewer's suggestion to consider environmental costs when determining the required recycling rate for imported plastic waste. While we recognize the importance of including all relevant costs - including import, operational, and

environmental - obtaining precise environmental cost data related to plastic waste import at the country level presents a challenge.

Therefore, we represented the minimum recycling rate of imported plastic waste (namely the required recycling rate) in this work, considering import costs, labour, rent, electricity, and physical losses as expenses. This rate already surpasses certain domestic or scenario-based recycling rates commonly used to simulate the fate of imported plastic waste.

We fully endorse the suggestion that if the data on environmental costs become available, incorporating such data would undoubtedly lead to a more accurate estimation of the recycling rate. In the final paragraph of the discussion section, we emphasize the required recycling rate could be even higher when considering environmental costs, capital investment, and other operational expenses. The added content (lines 243 to 248) is as follows:

“Our current approach of calculating the *RRR* focuses on basic cost factors to provide a minimum benchmark for recycling imported plastic waste. The actual *RRR* across countries could potentially be even higher when considering additional cost factors such as environmental costs, capital investment, and other operational expenses (e.g. chemical feedstocks²⁷, maintenance costs²⁸, and value-added taxes²⁹). Future research is encouraged to delve into these factors to better capture the recycling landscape.”

7) Line 239-251: Normally, for countries like China, there are significant differences in labor and rental costs between big and small cities. I hope this can be reflected in this article.

We acknowledge the concern that this approach may not account for regional variations in labour and rental costs within countries like China, where significant differences exist between big and small cities.

The rationale behind using labour costs at the country level assumes an equitable distribution of workers' earnings, rather than being concentrated solely in large cities with higher wages or in smaller cities and rural areas. This premise aligns with the actual distribution of plastic recycling industries across countries, which are typically

dispersed across various cities with access to plastic waste or situated in coastal cities that facilitate waste transport via ports.

Furthermore, our sensitivity analysis, as illustrated in Figure 4 (Figures S4-6 for other countries), indicates that fluctuations in labour costs have a comparatively minor impact on *RRR* compared to ‘product price’ (primary plastic) and ‘costs for waste imports’ across countries.

However, we agree that the regional disparities should be discussed and reflected in this work. Especially when drawing conclusions or making comparisons regarding the *RRR* across countries, it is essential to carefully reframe the applicable coverage and highlight the potential to delve into the regional disparity of the *RRR*. Therefore, we have revised the sentences in the main text as below:

In lines 137 to 138 in the results section, we have rephrased ‘labour cost is a key factor driving the difference between...’. Instead, we began with “Labour costs for recycling imported plastic waste in Asian countries are generally lower compared to those in European countries.”

In lines 149 to 151 in the results section, we have ended this paragraph describing the labour costs by including “It is worth noting that all costs were collected at the country level, which may not fully capture the disparities between countries and within countries due to regional differences.”

In lines 254 to 258 of the discussion section, we have underscored the limitation of accessing regional data in this work and emphasized the necessity for further investigation to address this gap:

“Moreover, though we credit this work for providing country-specific recycling rates for imported plastic waste, it is important to note that regional disparities within a country are not reflected, complicating the comparison of *RRR* among countries. Enhancing the resolution of *RRR* at both regional and city levels could provide a more nuanced understanding and better support for local policies.”

In lines 276 to 278 of the method section, we remind readers that relying on city or regional data for the required recycling rate (*RRR*) is promoted and applicable, as indicated by the provided equation:

“The cost-benefit equation is compatible with and encouraged for data collected at a higher resolution, such as city or regional levels, which could better reflect the *RRR* across geographical units.”

8) Line 293-301: Many previous studies have provided various methods for processing trade data. Like:

i. “Brewer, T. D. et al. A method for cleaning trade data for regional analysis: The Pacific Food Trade Database (version 2, 1995-2018). f6d6974d31f146110da3af7eee8f844f.pdf (windows.net)”

ii. “Chen. C. et al. Advancing UN Comtrade for Physical Trade Flow Analysis: Review of Data Quality Issues and Solutions. doi.org/10.1016/j.resconrec.2022.106526”

iii. “Szkutnik, T. et al. Identification of Outliers in High Density Areas with the Use of a Quantile Regression Model. CEEOL - Article Detail”

iv. Why does this study only refer to the data of the reporter country?

We appreciate the reviewer's insightful observation regarding trade data solely from reporting countries in our study. We agree that incorporating data from both reporting countries and their trading partners is essential for a comprehensive analysis.

In the UN Comtrade database, importers typically report trade values on a CIF basis, encompassing transport costs, whereas exporters often report values on an FOB basis, excluding transport costs. In calculating the *RRR*, we aim to incorporate transport costs associated with purchasing plastic waste while excluding those associated with selling primary plastics, as buyers shoulder these costs. Based on this, we prefer the trade value of plastic waste import and primary plastic export (both reported by the research countries), while the former trade value is CIF-based and the latter trade value is FOB-based.

However, relying solely on trade data reported by one side of trading countries may result in unreliable and less robust analyses, as the reviewer concerned. Consequently,

we integrated mirror trade data reported by the trading partners of the 22 research countries into our analysis. In Table S2, we have presented the comparative *RRR* results using both the trade data of the 22 research countries (plastic waste imports and primary plastics exports) and the data from their trading partners (plastic waste exports and primary plastics imports).

In addition, corresponding to Figures 2-4 in the main text, we have included supplementary figures with *RRR* results calculated using mirror trade data (Figure S1-3) in the supplementary file. This addition aims to enhance the transparency and accessibility of the results.

Furthermore, we have followed the reviewer's suggestion to remove trade data outliers. Referring to the valuable references the reviewer provided, we have conducted a distributional analysis of the value-to-mass ratio for all trade entries of specific plastic types each year. By identifying outliers at a distance greater than three standard deviations from the mean value, approximately 2.4% and 2.6% of plastic waste and primary plastic trade entries were flagged as outliers, respectively.

Lastly, we extend our gratitude to the reviewer for prompting this refinement in our methodology, which enhances the robustness of our research outcomes.

The revised content is detailed as follows:

In lines 322 to 338 of the method section, we have revised our description to provide a clearer explanation of our approach to accessing trade data:

“Each trade entry typically includes details such as reporting country, partner country, period, net weight, and trade value. Ideally, each trade flow should be reported by both importer and exporter during the same period, with closely aligned net weight and trade values. However, discrepancies often arise due to varying reporting conventions; exporters typically report trade values as Free On Board (FOB), while importers report them on a Cost for Insurance and Freight (CIF) basis.

For our analysis, we require trade values for a country's imported plastic waste and its exported primary plastics, derived from the UN Comtrade database. There are two options: using trade values reported by the research country, including plastic waste import (CIF basis) and primary plastics export (FOB basis), or using mirror trade values reported by the trading partner of the research country, including plastic waste export (FOB basis) and primary plastic import (CIF basis).

In calculating the RRR via the cost-benefit equation, we aim to incorporate the international transport cost for importing plastic waste while excluding the transport revenue for recycled primary plastics. Therefore, we prioritize using plastic waste imports (CIF basis) and primary plastics exports (FOB basis), both reported by the research countries. However, for a robustness check, we also include results based on mirror trade data reported by the trading partners of the 22 research countries in the supplementary file (Table S2).”

In lines 339 to 344 in the method section, we have added how we identified and removed the trade value outliers (stepwise changes in trade entries when processing the original trade data are now provided in Table S3):

“We detected trade value outliers through a distributional analysis of the value-to-mass ratio for all trade entries of specific plastic types each year^{35,36,37}. Since most of these unit price distributions follow a lognormal pattern, we transformed them into normal distributions by taking the natural logarithm ($\ln(\$/\text{kg})$)³⁸. By identifying outliers at a distance greater than three standard deviations from the mean value, approximately 2.4% and 2.6% of plastic waste and primary plastic trade entries were flagged as outliers, respectively.”

We have included Table S2 in the supplementary file for comparing the RRR results by country and plastic waste type between using trade data of the research countries and their trading partners:

Table S2. Comparison of the required recycling rate using trade data reported by 22 research countries and their trading partners (mirror trade data). The trade data from the 22 research countries include plastic waste imports and primary plastics exports,

while the mirror trade data include plastic waste exports and primary plastics imports reported by the trading partners of the 22 research countries. The required recycling rate for each country was averaged over the period 2013-2022.

Country	PE		PS		PVC		Others	
	22 countries	Trading partners	22 countries	Trading partners	22 countries	Trading partners	22 countries	Trading partners
Belgium	0.709	0.696	0.537	0.630	0.883	0.820	0.763	0.748
Canada	0.642	0.680	0.774	0.777	0.808	0.841	0.778	0.763
China	0.580	0.563	0.681	0.571	0.672	0.640	0.725	0.606
Hongkong (China)	0.518	0.614	0.424	0.448	0.518	0.526	0.515	0.573
Czechia	0.625	0.599	0.484	0.502	0.507	0.566	0.513	0.482
France	0.748	0.842	0.482	0.532	0.844	0.851	0.822	0.812
Germany	0.663	0.665	0.647	0.655	0.829	0.852	0.658	0.704
India	0.572	0.532	0.503	0.439	0.572	0.627	0.644	0.583
Indonesia	0.459	0.447	0.409	0.468	0.537	0.509	0.668	0.549
Italy	0.763	0.786	0.709	0.685	0.630	0.696	0.818	0.835
Malaysia	0.657	0.593	0.592	0.491	0.609	0.584	0.650	0.656
Mexico	0.466	0.557	0.507	0.514	0.563	0.665	0.660	0.610
Netherlands	0.618	0.710	0.676	0.712	0.833	0.856	0.781	0.748
Taiwan (China)	0.568	0.546	0.578	0.472	0.614	0.584	0.756	0.631
Republic of Korea	0.642	0.572	0.701	0.612	0.748	0.680	0.704	0.725
Slovenia	0.472	0.474	0.510	0.386	0.773	0.651	0.536	0.542
Spain	0.689	0.690	0.549	0.505	0.815	0.847	0.816	0.780
Thailand	0.432	0.410	0.461	0.404	0.665	0.604	0.509	0.509
Turkey	0.435	0.397	0.502	0.446	0.475	0.406	0.485	0.447
USA	0.813	0.695	0.627	0.562	0.885	0.825	0.744	0.715
UK	0.852	0.800	0.586	0.655	0.846	0.890	0.885	0.894
Vietnam	0.535	0.480	0.506	0.503	0.625	0.460	0.637	0.528

We have included supplementary figures (Figures S1-3), presenting the results calculated based on the trade values reported by the trading partners of the research countries (mirror trade data). These supplementary figures correspond to the main text figures (Figures 2-4). For further details, please refer to the supplementary file.

Reviewer #3 (Remarks to the Author):

The research perspective of the paper is unique. It tells an interesting fact that the results of the research have practical reference value for policy decision-making. The research method of the paper is reasonable, the data is sufficient, the argumentation process is reasonable and the conclusion is scientific. The thesis logic is more rigorous, the style of writing is more rigorous. The following aspects need to be further improved in the paper:

We are grateful for the reviewer's acknowledgement of the novelty of our work. We are dedicated to further refining our work to elevate its quality.

(1) the importance of the conclusion of the study is not fully demonstrated.

We appreciate the reviewer's reminder regarding how to appropriately conclude the results and avoid concluding the results out of the research scope. We have carefully rephrased the impact of our *RRR* results and removed the inappropriate forecast of the environmental consequences associated with the plastic waste trade as we did not conduct an actual analysis on this.

In the last sentence of the abstract section (lines 21-23), we have deleted the forecast about environmental impact and highlighted our contribution in providing country-specific *RRR* that could facilitate research and policy efforts measuring and mitigating the associated environmental impacts:

“The country-specific required recycling rates provided by this study could facilitate research and policy efforts aimed at quantifying and mitigating the environmental impact of plastic waste trade.”

In the discussion section (lines 208-210), we have revised the sentence to reflect on the relationship between recycling rates, recycled volume, and the measurement of environmental impacts associated with plastic waste trade:

“The divergent recycling rates across countries result in varied estimations of recycled volume within traded plastic waste, thereby complicating the measurement of environmental impacts linked to the global plastic waste trade.”

In the discussion section (lines 216-217), we have dropped the imprecise description of directly linking recycling rates with the environmental impacts of plastic waste trade. Instead, we have revised the sentence to imply that “Such variations in recycling rates can lead to fluctuation in the estimation of environmental impacts associated with plastic waste trade.”

(2) the value of the results is not reflected sufficiently

In response to the reviewer's comment, we have enhanced the accessibility of our results. We included a comparison of *RRR* results using data from both the 22 research countries and their trading partners in Table S2, allowing readers to easily access *RRR* values by country and plastic waste type. The Table S2 in the supplementary file is shown as follows:

Table S2. Comparison of the required recycling rate using trade data reported by 22 research countries and their trading partners (mirror trade data). The trade data from the 22 research countries include plastic waste imports and primary plastics exports, while the mirror trade data include plastic waste exports and primary plastics imports reported by the trading partners of the 22 research countries. The required recycling rate for each country was averaged over the period 2013-2022.

Country	PE		PS		PVC		Others	
	22 countries	Trading partners	22 countries	Trading partners	22 countries	Trading partners	22 countries	Trading partners
Belgium	0.709	0.696	0.537	0.630	0.883	0.820	0.763	0.748
Canada	0.642	0.680	0.774	0.777	0.808	0.841	0.778	0.763
China	0.580	0.563	0.681	0.571	0.672	0.640	0.725	0.606
Hongkong (China)	0.518	0.614	0.424	0.448	0.518	0.526	0.515	0.573
Czechia	0.625	0.599	0.484	0.502	0.507	0.566	0.513	0.482
France	0.748	0.842	0.482	0.532	0.844	0.851	0.822	0.812

Germany	0.663	0.665	0.647	0.655	0.829	0.852	0.658	0.704
India	0.572	0.532	0.503	0.439	0.572	0.627	0.644	0.583
Indonesia	0.459	0.447	0.409	0.468	0.537	0.509	0.668	0.549
Italy	0.763	0.786	0.709	0.685	0.630	0.696	0.818	0.835
Malaysia	0.657	0.593	0.592	0.491	0.609	0.584	0.650	0.656
Mexico	0.466	0.557	0.507	0.514	0.563	0.665	0.660	0.610
Netherlands	0.618	0.710	0.676	0.712	0.833	0.856	0.781	0.748
Taiwan (China)	0.568	0.546	0.578	0.472	0.614	0.584	0.756	0.631
Republic of Korea	0.642	0.572	0.701	0.612	0.748	0.680	0.704	0.725
Slovenia	0.472	0.474	0.510	0.386	0.773	0.651	0.536	0.542
Spain	0.689	0.690	0.549	0.505	0.815	0.847	0.816	0.780
Thailand	0.432	0.410	0.461	0.404	0.665	0.604	0.509	0.509
Turkey	0.435	0.397	0.502	0.446	0.475	0.406	0.485	0.447
USA	0.813	0.695	0.627	0.562	0.885	0.825	0.744	0.715
UK	0.852	0.800	0.586	0.655	0.846	0.890	0.885	0.894
Vietnam	0.535	0.480	0.506	0.503	0.625	0.460	0.637	0.528

In addition, we have updated our Python scripts used for calculating the results and made them available in a publicly accessible data repository on Zenodo (<https://zenodo.org/records/10996867>). Readers can now access these scripts to customize and adapt the calculations for their own use.

(3) the analysis of the results is not thorough enough, and some important factors are neglected, such as the management cost of the circular supply chain, the cost of international transportation, the social overflow of plastic waste.

We acknowledge the reviewer's concerns regarding the exclusion of additional costs in our analysis to derive the *RRR*. However, it is important to clarify that the *RRR* we provided serves as a minimum benchmark for recycling imported plastic waste, deduced through cost-benefit analysis due to the unavailability of physical measurement data for some countries.

Our intention in this work was not to precisely determine the real recycling rate, as physical measurement via mass balance would be more robust and reliable for that purpose. Instead, our focus was on providing consistent country-specific *RRRs*, which

offer a novel approach to capturing the recycling status of imported plastic waste across countries. This fills a gap in the current data landscape, which may not be adequately addressed by physical measurement alone.

We have refined the last paragraph in the discussion section to indicate potential opportunities for future research that readers may wish to explore further.

“Our current approach of calculating the *RRR* focuses on basic cost factors to provide a minimum benchmark for recycling imported plastic waste. The actual *RRR* across countries could potentially be even higher when considering additional cost factors such as environmental costs, capital investment, and other operational expenses (e.g. chemical feedstocks²⁷, maintenance costs²⁸, and value-added taxes²⁹). Future research is encouraged to delve into these factors to better capture the recycling landscape. Due to data availability constraints, we used the value of primary plastic exports as a proxy for recycled plastic revenue to maintain data consistency. However, this method may introduce inaccuracies due to potential variations in the quality of recycled plastics. Besides, advancements in recycling technologies (such as chemical, enzymatic, and solvent-based recycling) may already enable higher-valued final products^{30,31}, which may not be accurately captured within the current classification of primary plastics. Moreover, though we credit this work for providing country-specific recycling rates for imported plastic waste, it is important to note that regional disparities within a country are not reflected, complicating the comparison of *RRR* among countries. Enhancing the resolution of *RRR* at both regional and city levels could provide a more nuanced understanding and better support for local policies. Finally, prioritizing the use of the actual recycling rate determined through mass balance is recommended due to its superior accuracy, whenever physical measurement is feasible. However, the *RRR* provides added value as it addresses data gaps and deduces recycling rates where physical measurement may not be practical.”

(4) the scope of application of the conclusions needs to be given a clearer demonstration

We appreciate the reviewer's insight regarding the need for a clearer demonstration of the application scope of our results. To address this, we have enhanced the illustration of the results' applicability in two ways.

Firstly, we have emphasized that the *RRR* results can be effectively compared across countries, as all costs and benefits used in deducing the *RRR* were collected at the country level. However, we also highlight that this does not prevent its application at regional or city levels if such data become available.

Secondly, we have underscored the preference for accessing the recycling rate of imported plastic waste physically via mass balance, especially when physical data is accessible. In cases where physical data is not available, the deduced *RRR* could serve as a viable option.

The revisions addressing these two points are detailed as follows:

In lines 49 to 51 of the introduction section, we provided additional context by highlighting the absence of physically measured data:

“However, determining the recycling rate for imported plastic waste in receiving countries is hindered by measurement difficulties, leading existing studies to rely on assumed domestic or scenario-based recycling rates, often lacking robust data support.”

In lines 149 to 151 in the result section, we indicated the boundary for collected data used to deduce *RRR*:

“It is worth noting that all costs were collected at the country level, which may not fully capture the disparities between countries and within countries due to regional differences.”

Moreover, in the discussion section (lines 258 to 262), we have underscored the preference for physical data obtained through mass balance when available.

“Finally, prioritizing the use of the actual recycling rate determined through mass balance is recommended due to its superior accuracy, whenever physical measurement is feasible. However, the *RRR* provides added value as it addresses data gaps and deduces recycling rates where physical measurement may not be practical.”

In lines 254 to 258 of the discussion section, we have underscored the limitation of accessing regional data in this work and emphasized the possibility to extend the research boundary at the regional or city level:

“Moreover, though we credit this work for providing country-specific recycling rates for imported plastic waste, it is important to note that regional disparities within a country are not reflected, complicating the comparison of *RRR* among countries. Enhancing the resolution of *RRR* at both regional and city levels could provide a more nuanced understanding and better support for local policies.”

In lines 276 to 278 of the method section, we remind readers that extending city or regional data for the required recycling rate (*RRR*) is promoted and applicable, as indicated by the provided equation:

“The cost-benefit equation is compatible with and encouraged for data collected at a higher resolution, such as city or regional levels, which could better reflect the *RRR* across geographical units.”

(5) the innovation point is presented more clearly

We appreciate the ambiguity about the novelty of our work pointed out by the reviewer. We have refined our innovation of this work by providing a country-specific database of the required recycling rate of imported plastic waste that can be used for studies measuring and mitigating the environmental impact of plastic waste trade, instead of emphasizing the difference of *RRR* to the domestic recycling rate. We have clearly stated it in the revised abstract:

“The environmental impact of traded plastic waste hinges on how it is treated. Existing studies typically assume domestic or scenario-based recycling rates for managing imported plastic waste....The country-specific required recycling rates provided by this study could facilitate research and policy efforts aimed at quantifying and mitigating the environmental impact of plastic waste trade.”

Moreover, we have included a review of previous work in the introduction section (lines 48 to 61), highlighting the reliance on domestic or scenario-based recycling rates in current research and the need for comprehensive, transparent data to inform policy.

“While the global plastic recycling rate remains low, there is an implicit assumption that traded plastic is primarily recycled¹³. However, determining the recycling rate for imported plastic waste in receiving countries is hindered by measurement difficulties, leading existing studies to rely on assumed domestic or scenario-based recycling rates, often lacking robust data support. For instance, Wen et al. quantified the shift in the environmental impact of plastic waste imports from China to Southeast Asia after 2017, assuming plastic recycling rates equalled domestic rates among five Southeast Asian countries, ranging from 10% to 40%⁷. Bourtsalas et al. faced similar challenges when estimating the environmental impact of treating imported plastic waste in the USA, assuming widely varying recycling rates ranging from the domestic recycling rate at 8.7% to the scenario-based 50%¹⁴. Bishop et al. encountered a lack of official data on the fate of exported plastics from Europe, prompting them to adopt a broader range of recycling rates from 50% to 90%¹⁵. This reliance on domestic or scenario-based recycling rates underscores the critical need for comprehensive, transparent data to inform policy and research efforts effectively.”

(6) the discussion section adds some policy implications.

Thanks for the reviewer’s reminder to expand on the policy implications of our work. In response, we have included a discussion on the necessity of tracking waste flows between waste exporting and importing countries through the establishment of a robust Prior Informed Consent (PIC) procedure. While the PIC is already legally binding among members of the Basel Convention, its effectiveness varies by country. We emphasize the positive example set by OECD countries in implementing a transparent disclosure system for PIC. Additionally, we have addressed the potential to mitigate environmental impacts associated with plastic waste trade by directing exports to countries with well-established waste treatment facilities, such as those

within the EU and OECD. The added content in the discussion section (lines 233 to 242) is detailed as follows:

“Although the *RRR* may be a conservative estimate, it suggests that significant losses cannot be excluded. Ensuring transparent tracking systems for plastic waste exports is crucial, such as implementing a robust Prior Informed Consent procedure between waste importing and exporting countries²⁴. A notable example is the OECD control system for waste recovery, mandating disclosure of pre-consented recovery facilities and technology types in the waste importing country²⁵. Additionally, while recycling in developed countries may incur higher costs, it often results in a lower overall environmental impact compared to most waste-importing countries. These considerations carry weight in policymaking, as evidenced by recent proposals for plastic waste export bans within the EU²⁶.”

Reviewers' Comments:

Reviewer #1:

Remarks to the Author:

The authors have thoroughly addressed previous review comments, in particular adding methodological detail (including new Tables in the SI) and providing more robust contextualisation based on specific points drawn from recent references. I consider this manuscript to be suitable for publication in Nature Communications. The authors may wish to consider a few minor comments below.

L149: is 6-20% of cost really “minor”? Or is variation in these elec and rent costs minor relative to other factors?

L210-212: Implications for average recycling rate across the 22 studied countries are emphasised here – but it would be useful to highlight the spread of RRR across countries, and the value of these country-specific estimates for country-specific modelling studies...

L227 “than conventional modelling assumptions”?

Reviewer #2:

Remarks to the Author:

This manuscript has been greatly improved. There are still some issues that should be solved before acceptance. The comments are listed below:

Major comments:

- (1)Line 1-2: This title is not informative now. “Structural difference” may be confusing for readers because it does not appear or gain some explanations in the main text. And “revealed” seems to be needless.
- (2)Line 85: The assumption of the recyclate can be sold at a price close to primary plastics should be supported by some evidences. For example, the prices of primary and secondary plastics can be shown in supporting information.
- (3)Line 90: The other types of recycling costs that are not considered by this study should also be listed, such as equipment maintenance cost. And the reasons of neglecting these costs should be given in this study.
- (4)Line 108-116: The waste plastic recycling costs calculated by this study are recommended to be compared to the values of previous studies. It can help improve the reliability of this study.
- (5)Line 117-136: The grade of imported waste plastics may influence RRR. It would be beneficial to conduct a thorough investigation into the correlation between RRR and the grade of imported plastic waste in different countries.
- (6)Line 156-158: Will it be more reasonable to weigh the average values of recycling various types of plastic waste based on their import values from an economic perspective?
- (7)Line 207: The potential application scenarios of RRR developed by this study deserves an in-depth analysis because it can highlight the importance of this study.

(8)Line 208-224: The values of RRR are changing with many factors (incl., technology, policies, etc). The authors mention that China's 2018 ban on plastic waste imports will inevitably have a significant impact on the domestic recycling rate. Thus, it is recommended to discuss the dynamics of RRR in these countries. Or the authors can provide some evidences to prove that the RRR will not change significantly during the studied time period.

(9)Line 314: The transit trade of imported plastic waste should also be considered in this study. It may have a large influence on RRR.

(10)Line 353-381: Why not use constant prices for calculation when analyzing the recycling rate of imported plastic waste from an economic perspective?

(11)Line 369-370: There are significant differences in the prices of industry electricity in different regions of countries. The use of average electricity price may lead to a deviation on the calculation of recycling costs.

Minor comments:

(1)Line 28/37: Please add references.

(2)Line 69-71: The imported plastic wastes will not be fully dumped in landfills or incinerated because these imports have economic values. The statement of this sentence should be more rigorous.

(3)Table S1: Does the share of plastic types in plastic recycling here refer to imported recycling or domestic recycling? Can you provide data from earlier years for comparison with the present? Or give some evidence to indicate that there has been no significant change in this value during this period.

(4)Table S2: The same as Table S1, Since the annual RRR has been calculated, it is recommended to provide pictures or tables to make the average value more convincing.

(5)Table S9: Is the value in the "m²/tonne * yr" column a decimal point or a comma?

(6)Table S11: Physical loss by plastic types during the plastic waste mechanical recycling is also influenced by many factors such as technology. Can the author provide more relevant values or demonstrate that there was no significant change in this value during the research period?

Reviewer #3:

Remarks to the Author:

I am satisfied with the revised manuscript and recommend it for publication

Point-to-point response to the reviewers' comments:

Reviewer #1 (Remarks to the Author):

The authors have thoroughly addressed previous review comments, in particular adding methodological detail (including new Tables in the SI) and providing more robust contextualisation based on specific points drawn from recent references. I consider this manuscript to be suitable for publication in Nature Communications. The authors may wish to consider a few minor comments below.

We appreciate your positive feedback on our revised work. We are fully committed to incorporating your additional suggestions to further strengthen our manuscript.

L149: is 6-20% of cost really “minor”? Or is variation in these elec and rent costs minor relative to other factors?

We appreciate the reviewer's reminder regarding the ambiguity in our statement. Upon reflection, we recognize that emphasizing the minor variation in electricity and rent costs relative to labour costs is not essential in this context. Therefore, we have removed this sentence to ensure a consistent narrative throughout the paragraph.

L210-212: Implications for average recycling rate across the 22 studied countries are emphasised here – but it would be useful to highlight the spread of *RRR* across countries, and the value of these country-specific estimates for country-specific modelling studies...

We agree with the reviewer that discussing the variation in country-specific *RRR* values alongside the average *RRR* is important for country-specific modelling studies. To aid readers' understanding of *RRR* dynamics, we have provided annual country-specific *RRR* data in a supplementary file and illustrated these in the line charts in Fig. S12-S15.

We have cited and compared our findings to the recycling rates used in previous studies, such as Wen et al.'s evaluation of the environmental impacts of plastic waste trade (lines 214 to 223):

“Notably, the *RRR* averaged approximately 63% across 22 major importers and four plastic waste types from 2013 to 2022, significantly differing from the corresponding average domestic recycling rate of 23%. Moreover, the country-specific *RRR* values exceed those reported in previous studies based on the average domestic recycling rate. For instance, compared to Wen et al.’s findings (2021)⁸, which assumed a recycling rate of 38% for imported plastic waste in Malaysia in 2018, our study suggests a minimum required recycling rate of Malaysia to balance importing costs and benefits, averaging at 58% (PE and PVC), 62% (PS), and 64% (Others) during the period 2013-2022. This may suggest a potential underestimation of recycled plastic volumes in waste-importing countries.”

L227 “than conventional modelling assumptions”?

We thank the reviewer for the careful examination of the discussion section. We have revised this part to emphasize the potential applications of *RRR*, rather than highlighting the differences compared to the domestic average, which was discussed in the previous paragraph. The revised discussion is as follows (lines 234 to 241):

“*RRR* enhances the accuracy of modelling the fate and impacts of traded plastic waste, which is crucial for ongoing scientific research and policy implementation. When integrated with each country's waste treatment structure, *RRR* can provide insights into the environmental impacts of global plastic waste trade on importing nations and facilitate analysis of environmental responsibility associated with waste outsourcing. Moreover, the annual *RRR* data across countries and plastic waste types sheds light on how external events influence the global plastic waste trade. For instance, we observed a significant increase in most countries' *RRR* in 2020, potentially linked to the crude oil price drop during that period²⁵.”

Reviewer #2 (Remarks to the Author):

This manuscript has been greatly improved. There are still some issues that should be solved before acceptance. The comments are listed below:

Thank you for your time and efforts to help improve this paper. We appreciate the constructive feedback and are committed to addressing the remaining issues to enhance the manuscript further.

Major comments:

(1)Line 1-2: This title is not informative now. “Structural difference” may be confusing for readers because it does not appear or gain some explanations in the main text. And “revealed” seems to be needless.

We acknowledge the reviewer's concern regarding the clarity of the current title. In response, we have refined it to ‘Plastic waste exported to the global south has better recycling rates than often assumed’.

With this revised title, we aim to convey two messages: firstly, that recycling rates for imported plastic waste may be higher than commonly perceived, and secondly, that the recycling volume from plastic waste trade to the global south may be potentially underestimated. We believe this title better encapsulates the main idea of our work.

(2)Line 85: The assumption of the recyclate can be sold at a price close to primary plastics should be supported by some evidences. For example, the prices of primary and secondary plastics can be shown in supporting information.

Thanks to the reviewer for bringing attention to this aspect. Historically, secondary materials have been widely used by manufacturers as a (low-cost) substitute for primary plastics, leading to a correlation between the prices of secondary and primary plastics.

This correlation has been discussed in Hopewell et al.'s work and OECD's report. We have incorporated this understanding into the revised sentence (lines 85 to 87):

'Our analysis considers physical losses throughout the recycling process and assumes that the recyclate can be sold at a price correlated with that of primary plastic equivalents^{17, 18}.'

Moreover, we have incorporated a description in the methods section, quoting that the HS code selected for primary plastics in the UN Comtrade database includes not only virgin plastics but also secondary plastics in primary formats (e.g., pellets, flakes, etc.). This makes the unit price of primary plastic used to indicate recycling revenue more closely approach the unit price of secondary plastic in this study. The revised sentence in the methods section is as follows (lines 317 to 320):

'To derive a consistent unit price of recycled plastic, we used the trade data for plastics in primary forms (i.e. plastic pellets, flakes, etc.) recorded in the UN Comtrade database from 2013 to 2022³⁵, which consists of both virgin and secondary plastics.'

In addition, we have discussed the potential impact of presenting recycling revenue based on the price of primary plastics on the *RRR* results. While this approach may potentially underestimate the *RRR*, it aligns with our aim of establishing a minimum *RRR* benchmark in this study. Furthermore, it is noteworthy that even with this potential underestimation, the calculated *RRR* remains significantly higher than the domestic average recycling rate, which reinforces the central point of this study. The revised sentence in the discussion section is as follows (lines 263 to 270):

'Furthermore, due to data availability constraints, we used the value of primary plastic exports as a proxy for recycled plastic revenue to ensure data consistency. However, advancements in recycling technologies, such as chemical, enzymatic, and solvent-based recycling, may lead to higher-valued final products that are not accurately captured within the current classification of primary plastics^{31, 32}. While this approach may 'underestimate' the *RRR* (as illustrated in Eq. 1), it aligns with our objective of establishing a minimum *RRR* benchmark, as the calculated *RRR* already exceeds the corresponding domestic recycling rates.'

Finally, we have included data on the unit price of primary plastics by country, year, and plastic waste type, derived from both trade and mirror trade data, in an Excel file provided as supplementary data (further detailed in our response to the major comment (8)).

(3)Line 90: The other types of recycling costs that are not considered by this study should also be listed, such as equipment maintenance cost. And the reasons of neglecting these costs should be given in this study.

We appreciate the reviewer's suggestion to list additional recycling costs and provide reasons for their exclusion. We have cited the work of Uekert et al. and Larrain et al. regarding the techno-economic assessment of plastic mechanical recycling. Our study focuses on the primary costs: electricity, labour, and land, alongside plastic waste imports. The revised sentence in the introduction (lines 91 to 93) now reads:

‘...to derive importing cost and primary plastic values. In addition, we consider the other main plastic recycling costs including labour, electricity, and rental costs for real estate^{19, 20}.’

Moreover, in the methods section under 'Costs and physical loss,' we have outlined all potential costs associated with mechanical recycling (lines 376 to 381):

‘The complete costs from importing plastic waste to producing recycled plastics include plastic waste imports, operational costs (electricity, labour, land rent, water, fuel, transportation, maintenance), fixed asset investments (buildings, machinery, equipment), potential environmental costs, and taxes^{19, 20, 39, 40}. Based on the work of Uekert et al., Larrain et al., and Faraca et al.^{19, 20, 39}, we consider the top three costs consistently across research countries from 2013 to 2022: electricity, labour, and rent.’

Finally, we have also discussed the potential impact of excluding certain costs on the *RRR* results (lines 258 to 263):

‘Our current approach to calculating the *RRR* focuses on main cost factors, providing a minimum benchmark for recycling imported plastic waste. The actual *RRR* across countries could be higher when considering additional cost factors such as environmental costs, capital investment, and other operational expenses (e.g. chemical feedstocks¹⁹, maintenance costs³⁰, and value-added taxes²⁰). Future research is encouraged to delve into the full costs and benefits of imported plastic waste to better capture the *RRR* dynamics.’

(4)Line 108-116: The waste plastic recycling costs calculated by this study are recommended to be compared to the values of previous studies. It can help improve the reliability of this study.

We value the reviewer's suggestion to compare the recycling costs with other literature to enhance the reliability of our study. In response, we have compiled Table S14 from various sources, which illustrates the mechanical recycling cost per kilogram of plastic waste, listed by country and plastic waste type. Our study's recycling costs, including electricity, rent, and labour, fall within the cost ranges reported in these studies.

In the paragraph referred by the reviewer, we have added a description pointing to Table S14 (lines 118 to 119):

‘To provide a comprehensive comparison, mechanical recycling costs of plastic waste from various literature sources are listed in Table S14.’

The Table S14 in the supplementary file is as follows:

Table S14. Mechanical recycling costs of plastic waste from the literature.

Reference	Country	Data year	Plastic waste type	Recycling cost (per kg)
Gradus et al., 2017 ¹⁵⁹	Netherlands	2017-2019	-	0.269 Euro
Lase et al., 2023 ¹⁶⁰	Belgium	2018	PE	0.545 Euro
Larrain et al., 2021 ³⁰	European countries	2019	PP, PS, and PE	0.363 Euro (PP) 0.348 Euro (PS) 0.424 Euro (PE)
Kim et al., 2023 ¹⁶¹	Korea	2020	PVC	0.180-0.210 USD

Nikiema and Asiedu, 2022 ¹⁶²	-	literature review (2012-2021)	PP, PET, PVC	0.003-0.230 USD
da Cruz et al., 2014 ¹⁶³	Romania and Portugal	2010	-	0.186 USD (Romania) 0.355 USD (Portugal)
Genc et al., 2019 ¹⁶⁴	Turkey	2016	PE, PET, and PP	0.200-0.450 USD

(5)Line 117-136: *The grade of imported waste plastics may influence RRR. It would be beneficial to conduct a thorough investigation into the correlation between RRR and the grade of imported plastic waste in different countries.*

We appreciate the reviewer's insight into the potential influence of the grade of imported plastic waste on *RRR*. However, distinguishing the grade of imported plastic waste is challenging. A higher unit cost for imported plastic waste could indicate higher quality, but it could also result from different plastic waste types classified under the same HS code, given that there are only four HS codes for all plastic waste. Therefore, we included the following sentence in the results section (lines 139 to 142) to acknowledge this complexity and data limitation:

“Moreover, the grade of imported plastic waste of the same type may influence *RRR*, e.g. through importing costs, which unfortunately cannot be fully reflected by the current Harmonized System (HS) codes for plastic waste.”

Additionally, to better track and quantify the recyclability of traded plastic waste, it is necessary to extend the HS code system to incorporate more plastic waste types beyond the current four types. We included this recommendation in the discussion section (lines 270 to 274):

“Additionally, we acknowledge the limitations of the HS system in adequately covering all types of traded plastic waste. The current classification of four categories may not effectively track the quality of traded plastic waste, highlighting the need for expanded coverage to reflect better the diversity of plastic materials being traded.”

(6)Line 156-158: Will it be more reasonable to weigh the average values of recycling various types of plastic waste based on their import values from an economic perspective?

We appreciate the reviewer's comment on the methods for averaging the *RRR*. We believe that a country trading more of a specific type of plastic waste should be given more weight in the average *RRR* for that waste type. Our rationale for prioritizing trade volume over trade value in averaging the *RRR* is that the recycling rate is fundamentally based on physical mass measurements.

However, we agree with the reviewer that using a value-weighted average could also be a viable method. Consequently, we have included the results of both mass-weighted and value-weighted average *RRRs* in Table S15. The trade values (reported in USD in the UN Comtrade database) have been adjusted to constant 2022 prices using the USA's annual Consumer Price Index (CPI).

Table S15. Comparison of the average *RRR* by trade data source and weighting methods. The mass-weighted and value-weighted averages are calculated using the traded plastic waste mass and traded value, respectively. Trade values from 2013 to 2022 have been adjusted to constant 2022 prices using the yearly Consumer Price Index (CPI) of the USA from the World Bank¹⁶⁵.

Trade source	data	Weighting method for averaging RRR	Average RRR	Average RRR of plastic PE	Average RRR of plastic PS	Average RRR of plastic PVC	Average RRR of plastic Others
Research countries		Mass-weighted	63%	58%	56%	70%	65%
Research countries		Value-weighted	64%	60%	59%	71%	67%
Trading partners of research countries		Mass-weighted	58%	54%	52%	68%	59%
Trading partners of research countries		Value-weighted	59%	56%	53%	69%	60%

We have included a sentence in the Fig. 3 caption (lines 182 to 183) pointing to this comparison table:

‘A comparison of average *RRR* weighted by trade mass and trade value, using both trade and mirror trade data, is shown in Table S15.’

(7)Line 207: The potential application scenarios of RRR developed by this study deserves an in-depth analysis because it can highlight the importance of this study.

We take note of the reviewer's suggestion on elaborating *RRR* application scenarios. We have identified two key directions for applying *RRR*:

Firstly, *RRR* enables the quantification of the environmental impact of plastic waste trade. *RRR* indicates the share of recycling and, implicitly, the non-recycled share, which connects to plastic mismanagement. By matching this waste treatment structure with trade volumes, *RRR* allows for an examination of the environmental impact of imported plastic waste in a country and helps account for the responsibility of waste outsourcing.

Secondly, inspired by the reviewer's major comment (8), we emphasize that the temporal change of *RRR* in a country could reflect external impacts, such as oil price fluctuations.

We have incorporated these discussions in lines 237 to 244:

‘*RRR* enhances the accuracy of modelling the fate and impacts of traded plastic waste, which is crucial for ongoing scientific research and policy implementation. When integrated with each country's waste treatment structure, *RRR* can provide insights into the environmental impacts of global plastic waste trade on importing nations and facilitate analysis of environmental responsibility associated with waste outsourcing. Moreover, the annual *RRR* data across countries and plastic waste types sheds light on how external events influence the global plastic waste trade. For instance, we observed a significant increase in most countries' *RRR* in 2020, potentially linked to the crude oil price drop during that period²⁵.’

(8)Line 208-224: The values of RRR are changing with many factors (incl., technology, policies, etc). The authors mention that China's 2018 ban on plastic waste imports will inevitably have a significant impact on the domestic recycling rate. Thus, it is recommended to discuss the

dynamics of RRR in these countries. Or the authors can provide some evidences to prove that the RRR will not change significantly during the studied time period

We are grateful for the reviewer's suggestion about the temporal changes in *RRR*. To better represent the *RRR* dynamics in this work, we have elaborated the annual *RRR* across countries and plastic types in the following ways:

Firstly, to visualize the temporal changes in *RRR* by country and plastic type, we have included line charts of annual *RRR* (including mirror *RRR*) across countries and plastic types in Figures S12-S15.

Figure S12 for waste PE is illustrated as follows (see Fig. S13-S15 in the supplementary file):

Fig.S12 Annual *RRR* of plastic waste PE from 2013 to 2022 in 22 research countries. The *RRR* and mirror *RRR*, calculated using trade data (plastic waste imports and primary plastics exports reported by the 22 research countries) and mirror trade data (plastic waste exports and primary plastics imports reported by the trading partners of the 22 research countries), are shown in purple and orange colours, respectively. Missing annual *RRR* values were linearly interpolated and are shown in grey.

Secondly, to facilitate readers' access to and analysis of *RRR* dynamics, we have compiled an Excel file containing annual *RRR* data across countries and types. This file also includes associated annual costs for waste import, labour, electricity, rent, and plastic recycling benefits. The Excel file is provided as an independent supplementary data file.

	A	B	C	D	E	F	G	H	I	J	K	L
1	country	period	waste imports	waste exports	labour	electricity	rent	primary plastics imp	primary plastics exp	RRR	RRR_mirror	type
2	Belgium	2013	0.67336006	0.67336006	0.231720013	0.074515264	0.040567418	2.127760188	2.202344586	0.551448433	0.57077836	PS
3	Belgium	2014	0.515109645	0.785663171	0.238784913	0.073460736	0.040579636	2.139086416	2.138488777	0.483171051	0.63360853	PS
4	Belgium	2015	0.40620958	0.537905081	0.199781763	0.061492928	0.033890182	1.53091466	1.60196495	0.52121589	0.647815303	PS
5	Belgium	2016	0.401298617	0.478970169	0.200144575	0.064867292	0.034414527	1.538919248	1.547128727	0.539190067	0.602151527	PS
6	Belgium	2017	0.483076838	0.576019274	0.208088825	0.063865707	0.0351234	1.707715372	1.732283779	0.543017519	0.615621441	PS
7	Belgium	2018	0.464942347	0.708936242	0.22223045	0.067723264	0.038650909	1.847180945	1.850758205	0.510438788	0.668677154	PS
8	Belgium	2019	0.246273885	0.409442509	0.382597338	0.066226635	0.038470091	1.471526225	1.498145951	0.582917289	0.725466879	PS
9	Belgium	2020	0.27353454	0.297480078	0.392018638	0.069055889	0.039250145	1.283413929	1.361900272	0.67645259	0.740032142	PS
10	Belgium	2021	0.380847202	0.428156311	0.411453025	0.080638063	0.041932091	2.088148632	2.115018958	0.514951132	0.548548966	PS
11	Belgium	2022	0.672543091	0.604930783	0.411453025	0.117082364	0.043077273	2.553136012	3.276303662	0.452075863	0.548598648	PS
12	Canada	2013	1.108195489	1.14409521	0.274678058	0.034562049	0.028023637	2.221375671	2.184090584	0.787872451	0.793887584	PS
13	Canada	2014	1.118746086	1.152931512	0.264243673	0.030718831	0.02897735	2.245552821	2.21045742	0.77698093	0.782960962	PS
14	Canada	2015	0.956160471	0.993675482	0.233800413	0.028159785	0.025217038	1.818637051	1.77743342	0.832753519	0.838443637	PS
15	Canada	2016	0.821468577	0.850576161	0.227959768	0.037119015	0.02553389	1.609965915	1.584786343	0.83538469	0.843842793	PS
16	Canada	2017	0.835081613	0.862269609	0.197648183	0.038400401	0.027042994	1.850037825	1.823458315	0.716961295	0.724155841	PS
17	Canada	2018	1.119395911	1.16009197	0.20366573	0.03882483	0.030002333	1.928522692	1.928522692	0.859212333	0.884333992	PS
18	Canada	2019	0.928814274	0.963707351	0.20585039	0.042668772	0.035184943	1.635920161	1.635920161	0.882362291	0.907754347	PS
19	Canada	2020	0.729854506	0.756426084	0.212586998	0.042668772	0.038882628	1.358639086	1.313702663	0.927941464	0.920532918	PS
20	Canada	2021	0.56903216	0.588879416	0.227998698	0.042668772	0.044666249	2.177117264	2.09960765	0.501434886	0.494435567	PS
21	Canada	2022	1.000996381	1.050748887	0.233368175	0.042668772	0.055842321	2.831805171	2.579285635	0.615192324	0.581249697	PS
22	China	2013	0.67336006	0.67336006	0.040394063	0.045119984	0.022079372	1.988702365	1.934859708	0.480503325	0.46749405	PS
23	China	2014	0.663063813	0.663063813	0.045339375	0.045715704	0.023237328	1.818547434	1.740798022	0.531609102	0.50888091	PS
24	China	2015	0.812727729	0.522131738	0.048860625	0.043445535	0.02395484	1.467213764	1.285841275	0.860089798	0.517982705	PS
25	China	2016	0.483568419	0.368695384	0.04951875	0.041749316	0.023100117	1.326215727	1.373491908	0.518262455	0.433621517	PS
26	China	2017	0.649892467	0.399149007	0.052005	0.03791454	0.024559291	1.513898629	1.350041056	0.674028265	0.403898715	PS
27	China	2018	0.834899858	0.594266705	0.058186875	0.038083754	0.027333616	1.659608351	1.478890209	0.771576082	0.514945754	PS
28	China	2019	0.692418691	0.692418691	0.059775938	0.038083754	0.027227729	1.330056625	1.174564849	0.828580527	0.731714382	PS
29	China	2020	0.716125621	0.716125621	0.065561125	0.038083754	0.028422576	1.086708877	0.932357941		0.929185205	PS
30	China	2021	0.73704563	0.73704563	0.077466563	0.038083754	0.033770319	1.664275074	1.426106155	0.73991542	0.634028564	PS
31	China	2022	0.823141704	0.823141704	0.077466563	0.028503986	0.032406222	2.009291908	1.572112339	0.72810627	0.569685692	PS
32	China, Ho	2013	0.586583842	0.536356928	0.092147388	0.058873858	0.029388976	1.855080896	1.925063524	0.474315865	0.459976828	PS
33	China, Ho	2014	0.544710934	0.494735675	0.099152625	0.060983067	0.03014051	1.865518139	1.816085661	0.481797034	0.437138783	PS
34	China, Ho	2015	0.444916121	0.420859932	0.103312875	0.062872661	0.030892169	1.619041823	1.640012559	0.466019824	0.454367537	PS
35	China, Ho	2016	0.351941529	0.347679352	0.1079988	0.062229528	0.031557163	1.522089262	1.547239987	0.426048215	0.429754568	PS
36	China, Ho	2017	0.354249008	0.370827497	0.111027613	0.061983572	0.032171299	1.769373595	1.70846116	0.389818561	0.387553069	PS
37	China, Ho	2018	0.383890726	0.326999661	0.11590865	0.064052905	0.032708619	1.294333973	1.944320602	0.365264631	0.49636655	PS
38	China, Ho	2019	0.368475691	0.332220263	0.1220813	0.064689546	0.033462621	1.425496554	1.625382152	0.431187358	0.461371169	PS

While we have not observed obvious changes in countries' *RRR* around 2017 and 2018 due to China's ban, we did notice an increase in *RRR* in 2020, which may be connected to the drop in oil prices that year (more waste needs to be recycled to balance the cost as recycling benefits decreased). Therefore, to encourage readers to explore the potential socio-economic impacts on plastic waste imports through the observation of *RRR* dynamics, we have included the following discussion (lines 241 to 244): ‘Moreover, the annual *RRR* data across countries and plastic waste types sheds light on how external events influence the global plastic waste trade. For instance, we observed a significant increase in most countries' *RRR* in 2020, potentially linked to the crude oil price drop during that period.’²⁵

(9)Line 314: The transit trade of imported plastic waste should also be considered in this study. It may have a large influence on RRR.

We value the reviewer's insight on the issue of transit trade. While the transit trade adds complexity to analysing a country's actual recycling flows, we believe its impact on calculating *RRR* is relatively minor. This is because we derive *RRR* using the unit price of traded plastic waste and primary plastics, rather than relying solely on the country's absolute trade volume or trade value, which may be influenced by transit trade.

However, we agree with the reviewer that readers should interpret *RRR* values carefully in countries involved in transit trade. We have emphasized in the discussion that readers need to be cautious when applying *RRR* values to specific contexts, such as quantifying the total recycled volume and related environmental impacts for transit regions like Hong Kong (China), where absolute trade volumes might be affected by transit activities. The relevant discussion is as follows (lines 279 to 282): *‘Moreover, we suggest carefully applying the RRR when quantifying the recycled volume and related environmental impact of plastic waste imports in transit countries or regions, such as Hong Kong (China), as the absolute imported trade volume may not reflect the actual situation.’*

(10)Line 353-381: Why not use constant prices for calculation when analyzing the recycling rate of imported plastic waste from an economic perspective?

Thanks for the reviewer's suggestion regarding the use of constant prices when analysing *RRR*. We opted not to use constant prices because the costs (waste import, electricity, labour, rent) and the recycling revenue (primary plastics) used to calculate *RRR* were collected annually at the country level for each research year from 2013 to 2022 and converted to USD. This approach ensures that the deduced *RRR* maintains consistency in terms of period dimensions for all costs and benefits.

However, we appreciate the reviewer's reminder about constant prices. Building on this insight, we have included a value-weighted average *RRR* by adjusting the country's trade value of imported plastic waste (in USD) to constant 2022 prices using the USA's yearly consumer price index, as addressed in the major comment (6).

(11)Line 369-370: *There are significant differences in the prices of industry electricity in different regions of countries. The use of average electricity price may lead to a deviation on the calculation of recycling costs.*

We appreciate the reviewer's concern regarding the use of a uniform electricity price within a country. This concern extends to other cost components in deducing *RRR*, such as regional variations in labour and rent costs, which may better reflect *RRR* than uniform national values. The data barrier is that all the costs and recycling benefits need to be scaled to the same regional level when calculating a regional *RRR*. This also brings challenges as the waste importing cost and the primary plastics price collected from UN Comtrade are presented by national data only.

In this study, one of our goals is to reveal the national *RRR* difference among the top waste-importing countries. While we acknowledge that this national *RRR* may not accurately reflect regional differences in electricity, rent, labour, etc., within a country, it does not hinder a consistent comparison of *RRR* among the 22 research countries. This is because data are uniformly collected or aggregated across these countries, ensuring a reliable basis for comparison.

To reflect this issue in the revised manuscript, we have made two key points. Firstly, we have acknowledged that the calculated country-specific *RRR* may not fully capture regional differences within a country due to varied costs and recycling benefits across regions. We have therefore proposed a future research direction for *RRR* estimation at a local scale. The revised sentences are as follows (lines 274 to 279 in the last paragraph of the discussion section):

‘While our work provides valuable insights into country-specific recycling rates for imported plastic waste, it is important to recognize that regional disparities within a country are not accounted for. Variations in costs such as electricity, labour, and rent may vary significantly across regions, underscoring the need for future endeavours to derive *RRR* at both regional and city levels to support more localized policymaking.’

Secondly, we have suggested readers use the provided calculation framework to explore regional *RRR* by collecting regional cost and recycling benefit data (lines 299 to 301 in the Methods section):

‘The cost-benefit equation is compatible with and encourages data collected at city or regional levels, such as regional costs for electricity, labour, rent, etc., which could better reflect the *RRR* across geographical units.’

Minor comments:

(1)Line 28/37: Please add references.

We appreciate the reviewer's academic rigour. We have included the relevant reference associated with the given value in the sentence. The reference is shown as follows:

4. Our World in Data. Ocean plastics: How much do rich countries contribute by shipping their waste overseas? <https://ourworldindata.org/plastic-waste-trade> (2022).

(2)Line 69-71: The imported plastic wastes will not be fully dumped in landfills or incinerated because these imports have economic values. The statement of this sentence should be more rigorous.

We thank the reviewer for the careful examination. We have rephased the sentences below to make it more rigorous (lines 69 to 73): 'If such imports were not processed into recyclates that could be sold for a positive value—i.e. if the imports were primarily dumped or burned—the importing companies would incur irrecoverable losses. No company importing plastic waste could survive such a situation in the long term.'

(3)Table S1: Does the share of plastic types in plastic recycling here refer to imported recycling or domestic recycling? Can you provide data from earlier years for comparison with the present? Or give some evidence to indicate that there has been no significant change in this value during this period.

Thanks to the reviewer for the valuable feedback on Table S1. As outlined in the Method section, our approach extends the recycled output of plastic waste PE to HDPE and LDPE, and PET and PP for plastic waste 'Others', while considering variations in *RRR* resulting from recycled product pricing.

We have clarified in the figure caption of Table S1 that the recycling share refers to the country's overall plastic waste, including both domestically generated and imported plastic waste. Although we recognize that deriving the recycling share solely from imported plastic waste would provide more precise insights, the current HS code system

does not allow for such differentiation, and consistent data across research countries is lacking.

Moreover, as the reviewer rightly implied, the recycling structure may evolve over the research period (2013-2022). Due to limitations in data availability, obtaining annual data to track these changes is challenging. To address this limitation, we have conducted sensitivity tests on *RRR* results for waste PE and 'Others' by assuming exclusive recycling of PE waste into either HDPE or LDPE and of plastic waste 'Others' into either PET or PP. This allowed us to assess the impact of potential changes in the recycled product structure on *RRR* values. The resulting *RRR* values for PE and 'Others' were subjected to average $\pm 5\%$ and $\pm 10\%$ variations, respectively.

We have incorporated this clarification into the table caption to enhance readers' understanding. Additionally, detailed variations in *RRR* across countries and years, as influenced by the recycled product structure, are provided in the '*RRR* changes by product range' sheet within the supplementary data file.

The revised caption for Table S1 is as follows:

Table S1. Estimated share of recycled HDPE and LDPE from recycling PE waste, and PET and PP from recycling plastic waste 'Others'. Regional averages are used when data is unavailable in the research country, with regions defined as in Table S12. The recycling share is related to the overall plastic waste of a country, including both domestically generated and imported plastic waste. Due to incomplete data availability for each research year, we tested the sensitivity of *RRR* results by assuming recycling PE waste into either HDPE or LDPE exclusively, and recycling plastic waste 'Others' into either PET or PP exclusively. The resulting *RRR* values for PE and 'Others' were subject to average $\pm 5\%$ and $\pm 10\%$ variations, respectively. For detailed results by country and year, please refer to the '*RRR* changes by product range' sheet in the supplementary data.

Country	Share of PET recycling	Share of PP recycling	Share of HDPE recycling	Share of LDPE recycling
Malaysia ¹	29 (2019)	5 (2019)	52 (2019)	8 (2019)
Thailand ²	15 (2018)	19 (2018)	30 (2018)	26 (2018)
Philippines ³	14 (2019)	22 (2019)	6 (2019)	11 (2019)
Vietnam (Hanoi) ⁴	42 (2022)	-	30 (2022)	1 (2022)
Indonesia (Greater Jakarta) ⁴	65 (2020)	11 (2020)	11 (2020)	5 (2020)

Indonesia (Makassar) ⁴	70 (2020)	10 (2020)	10 (2020)	5 (2020)
India ⁴ (Delhi)	14 (2022)	7 (2022)	20 (2022)	25 (2022)
India ⁴ (Mumbai)	11 (2022)	4 (2022)	22 (2022)	26 (2022)
India ⁴ (Chennai)	15 (2022)	5 (2022)	20 (2022)	27 (2022)
Hongkong, China ⁵	3 (2020)	6 (2020)	10 (2020)	9 (2020)
Taiwan, China ⁶	55 (2019)	21 (2019)	-	-
China ⁷	33 (2022)	21 (2022)	10 (2022)	12 (2022)
Japan ⁸	30 (2021)	23 (2021)	20 (PE in total; 2021)	20 (PE in total; 2021)
United Kingdom ⁹	34 (2017)	7 (2017)	26 (2017)	3 (2017)
United States ¹⁰	31 (2017)	2 (2017)	20 (2017)	11 (2017)
Spain ¹¹	22 (2011)	4 (2011)	24 (2011)	29 (2011)
Germany ¹²	13 (2017)	26 (2017)	22 (2017)	20 (2017)
EU27 ¹³	22 (2019)	16 (2019)	18 (2019)	21 (2019)

The detailed variations in *RRR* across countries and years, influenced by the recycled product range, are provided in the '*RRR* changes by product range' sheet within the supplementary data file. Here is a screenshot to illustrate these variations (for column name explanations, refer to the 'note' sheet):

	A	B	C	D	E	F	G	H	I	J
1			group	trade			mirror trade			
2			subgroup	rate	change_m	change_m	rate_mirror	or_change	or_change	max (%)
3	type	country	period							
4	PE	Belgium	2013	0.584655	-0.78277	0.928992	0.54133	-2.30933	2.836128	
5			2014	0.633323	-2.56655	3.170619	0.592924	-2.79259	3.467841	
6			2015	0.650087	-2.35794	2.899035	0.610474	-4.71144	6.121573	
7			2016	0.691591	-1.66455	2.014637	0.659598	-4.77336	6.211307	
8			2017	0.65602	-2.23085	2.734841	0.650619	-3.08563	3.857822	
9			2018	0.635786	-1.37826	1.212397	0.676824	-0.65378	0.773697	
10			2019	0.800394	-0.62602	0.542898	0.919894	-0.28985	0.340295	
11			2020	0.951466	-2.23979	2.00314	0.866466	-3.06636	3.832007	
12			2021	0.746616	-0.28727	0.337243	0.674617	-7.77057	10.90098	
13			2022	0.744938	-3.91725	4.993977	0.764434	-4.49623	5.811773	
14		Canada	2013	0.71958	-5.01767	2.992455	0.744937	-5.39513	3.238105	
15			2014	0.572258	-4.99155	2.975568	0.583905	-5.26525	3.153226	
16			2015	0.440219	-5.38885	3.233995	0.451882	-6.13596	3.729482	
17			2016	0.619032	-7.01561	4.32937	0.621398	-6.63986	4.070891	
18			2017	0.682917	-5.03348	3.002679	0.699779	-3.97004	2.326698	
19			2018	0.672021	-4.93697	2.940335	0.681988	-4.57284	2.706923	
20			2019	0.759949	-3.2735	1.896662	0.805384	-3.22108	1.864692	
21			2020	0.68535	-6.85688	4.219768	0.772459	-6.80741	4.185731	
22			2021	0.65025	-4.10065	2.408441	0.752261	-3.03927	1.754239	
23			2022	0.623313	-3.74067	2.183996	0.687607	-3.37047	1.955944	
24		China	2013	0.657746	-0.93766	1.148895	0.416628	-1.63117	2.030264	
25			2014	0.625049	-1.92752	2.41545	0.421364	-2.74381	3.504097	
26	2015		0.587369	-4.04528	5.328555	0.447809	-2.44822	3.105106		
27	2016		0.462719	-2.07388	2.607636	0.425147	-2.37358	3.005223		
28	2017		0.404178	-4.96595	6.690028	0.421746	-1.95922	2.456971		
29	2018		0.547505	-3.64309	4.752618	0.432486	-4.45441	4.042105		
30	2019		0.543331	-7.15358	10.1876	0.713717	-1.17646	1.449261		
31	2020		0.590538	-18.346	36.91422	0.78609	-6.18623	8.592947		
32	2021		0.704764	-17.0368	32.70061	0.774729	-5.94714	8.21085		
33	2022		0.677168	-16.8056	31.99661	0.785368	-3.31188	4.286582		
34	Hongkong (Ch	2013	0.454811	-2.74278	2.604218	0.373387	-8.57437	11.63278		
35		2014	0.452935	-2.52495	2.963464	0.425307	-9.34629	12.93747		
36		2015	0.498155	-3.00336	3.56297	0.485913	-4.34573	5.316331		
37		2016	0.483047	-1.86861	2.161505	0.437419	-1.46551	1.356733		
38		2017	0.495205	-4.18841	5.105207	0.605167	-9.66164	10.65062		

(4)Table S2: The same as Table S1, Since the annual RRR has been calculated, it is recommended to provide pictures or tables to make the average value more convincing.

We value the review’s suggestion to provide readers with more information regarding the annual RRR. As addressed in the major comment (8), we have taken steps to address this concern by compiling an Excel file containing detailed annual RRR data across countries and types, which is included in the supplementary data file. This Excel file not only presents the annual RRR values but also includes associated costs for waste import, labour, electricity, rent, and plastic recycling benefits. Furthermore, to visually illustrate the annual RRR changes, we have created Figures S12-S15. Both screenshots of the Excel file and Figures S12-15 are provided in our response to the major comment (8).

*(5)Table S9: Is the value in the "m2/tonne * yr" column a decimal point or a comma?*

Thanks for the reviewer's reminder. The values shown in "m2/tonne * yr" should use a decimal point instead of a comma. We have made the necessary corrections.

(6)Table S11: Physical loss by plastic types during the plastic waste mechanical recycling is also influenced by many factors such as technology. Can the author provide more relevant values or demonstrate that there was no significant change in this value during the research period?

We agree with the reviewer and acknowledge that advanced plastic recycling methods such as chemical, enzymatic, and solvent-based recycling, may evolve over time. In contrast, mechanical recycling is a traditional and widely used method that relies on the physical properties of materials. Therefore, we believe the physical loss in mechanical recycling remains relatively stable despite technological advancements. We have included this perspective in the table caption, citing Jin et al. and Oblak et al.'s work on plastic mechanical recycling.

However, we understand the reviewer's concern about the assumption of constant physical loss among plastic waste types during the research period. We have addressed this by incorporating the uncertainty of physical loss into our sensitivity analysis. In this analysis, the values for specific plastic waste types vary among countries with different levels of development, as shown in Table S11. Although this approach does not explicitly account for temporal changes in physical loss within individual countries, it allows us to analyze physical loss across countries, which may reflect the impact of technological evolution over time. The detailed sensitivity analysis of *RRR*, including the parameter physical loss by country and plastic waste type, is provided in in Fig. 4 and Figs. S3-S9, including *RRR* calculated using both trade data and mirror trade data.

The revised Table S11 is shown below:

Table S11. Physical loss by type during plastic waste mechanical recycling (%). We assume the physical loss for each plastic type remained constant during the research period due to the stable physical properties observed in mechanical recycling processes^{118, 119}. Furthermore, the sensitivity analysis considering the impact of physical loss uncertainty on *RRR* values is provided in Fig. 4 and Figs. S3-S9.

References	Coverage	HDPE	LDPE	PS	PVC	PET	PP
Arena, Mastellone and Perugini ¹²⁰	Global	0.24	0.24	0.24	0.24	0.24	0.24
Civancik-Uslu, Nhu ²⁶	Belgium	0.19	0.19	0.11	-	-	0.15
Brouwer, Picuno ¹²¹	Netherlands (separate collection)	0.08	0.08	-	-	0.12	0.24
Brouwer, Picuno ¹²¹	Netherlands (MSW)	0.20	0.20	-	-	0.15	0.27
Faraca and Astrup ¹²²	Denmark	-	-	-	0.12	-	-
Plinke, Wenk ¹²³	Europe	-	-	-	0.10	-	-
Larrain, Van Passel ³⁰	Belgium	-	0.2	0.13	-	-	0.13
Uekert, Singh ²⁹	USA	0.17	0.28	-	-	0.26	0.17

Please see Fig. 4 in the manuscript and Fig. S3-S9 in the supplementary file.

Reviewer #3 (Remarks to the Author):

I am satisfied with the revised manuscript and recommend it for publication

We are pleased to hear that the reviewer is satisfied with our revisions. Thank you for your efforts in collectively improving the quality of this work.

Reviewers' Comments:

Reviewer #1:

Remarks to the Author:

The authors have further improved the manuscript, and have addressed the minor suggestions I had made during the last round of reviewing. My only remaining minor comment is that the new supplementary figures S12-S15 should be cited within the main manuscript. In my view this manuscript is ready for publication in Nature Communications.

Reviewer #2:

Remarks to the Author:

This manuscript has been greatly improved but some minor revisions should be made before accept. The detailed comments are listed below.

1. Line 1-2: The revised title 'Plastic waste exported to the global south has better recycling rates than often assumed' seems not to be very suitable because it cannot sufficiently reflect the innovation of this study. I think the authors developed an indicator of RRR representing the economic recycling requirements of waste plastic. Thus, I recommend the authors to refine the title.
2. Line 19: "23%" is a little ambiguous. The readers may think that 23% is the minimum required recycling rate concluded by previous studies.
3. Line 11-23: This abstract did not include sufficient important results of this study. It seems a little not informative.
4. Line 118-119: There are some differences in the costs of waste plastic recycling between mechanical recycling and chemical recycling. Here, the authors highlighted the costs of mechanical recycling. The potential impacts of different recycling technologies on the recycling costs should also be discussed.
5. Line 140-142: I understand that it is very difficult to get imported waste plastic grade information from HS trade data. The authors should give a more detailed discussion about that such as the import or export standards of waste plastics of different countries. And this issue should be considered in future studies.
6. Line 376-381: I did not find the specific costs or cost ranks of waste plastic recycling from the references 19, 20, and 40, which makes me cannot confirm whether the top three costs are electricity, labor, and rent. Please provide the specific positions of costs in these references. Or recheck these references.
7. Line 237-244: I think another important application of RRR is to help one country to formulate the recycling targets of waste plastic, which should be added in the main text. And RRR cannot reflect the environmental impacts of waste plastic trade.
8. Line 243-244: I did not understand why an increased oil price will promote an increase in RRR although you have given a short explanation.

REVIEWERS' COMMENTS

Reviewer #1 (Remarks to the Author):

The authors have further improved the manuscript, and have addressed the minor suggestions I had made during the last round of reviewing. My only remaining minor comment is that the new supplementary figures S12-S15 should be cited within the main manuscript. In my view this manuscript is ready for publication in Nature Communications.

We have cited the new supplementary figures S12–S15 in the main text (lines 555) as per the reviewer's suggestion. We appreciate the reviewer's thoughtful comments and patient guidance throughout the review process. Thank you for helping us improve our manuscript.

Reviewer #2 (Remarks to the Author):

This manuscript has been greatly improved but some minor revisions should be made before accept. The detailed comments are listed below.

1. Line 1-2: The revised title 'Plastic waste exported to the global south has better recycling rates than often assumed' seems not to be very suitable because it cannot sufficiently reflect the innovation of this study. I think the authors developed an indicator of RRR representing the economic recycling requirements of waste plastic. Thus, I recommend the authors to refine the title.

We appreciate the reviewer's insightful suggestion regarding the title. We have revised it to: “Economic viability requires higher recycling rates for imported plastic waste than expected.” This new title more accurately reflects the key findings and contributions of our study.

2. Line 19: “23%” is a little ambiguous. The readers may think that 23% is the minimum required recycling rate concluded by previous studies.

We understand that the ambiguity might arise from the word 'reported,' which could be misconstrued as referencing previous studies. We have rephrased the sentence to remove 'reported' from 'average reported domestic recycling rate' to clarify the meaning.

3. Line 11-23: This abstract did not include sufficient important results of this study. It seems a little not informative.

We believe the main contribution of this paper is the recycling rate data for imported plastic waste (*RRR*). Due to word limitations in the abstract, we focused on summarizing a general *RRR* of 63% and comparing it with the average domestic rate of 23%. We highlight the environmental consequences resulting from these disparate rates and emphasize the significance of the data provided, specifically through the lens of the *RRR*. We aim to attract readers by showcasing the recycling rate gap and its environmental impacts, encouraging them to explore the detailed data and findings in the full study.

4. Line 118-119: There are some differences in the costs of waste plastic recycling between mechanical recycling and chemical recycling. Here, the authors highlighted the costs of mechanical recycling. The potential impacts of different recycling technologies on the recycling costs should also be discussed.

We appreciate the reviewer's attention to this detail. We've clarified in the last paragraph of the introduction that the term 'recycling' in this paper refers specifically to mechanical recycling, which is the predominant method used in Global South countries. We also addressed the potential impact of other recycling technologies on costs and the required recycling rate (*RRR*).

Clarification in the introduction (Lines 81 to 82):

'Here, "recycling" specifically refers to mechanical recycling, the predominant method for recycling imported waste in Global South countries^{22, 23}.'

Discussion on different technologies (Lines 220 to 226):

“Due to data constraints, we used primary plastic exports as a proxy for recycled plastic revenue to ensure consistency. However, advancements in recycling technologies (e.g., chemical, enzymatic, and solvent-based methods) may create higher-value products not captured by current primary plastic classifications^{33, 34}, potentially leading to an overestimation of the *RRR* in some developed countries. Additionally, the four HS codes under 3915 may not fully reflect the quality and diversity of plastic waste, indicating a need for expanded classification coverage.”

5. Line 140-142: I understand that it is very difficult to get imported waste plastic grade information from HS trade data. The authors should give a more detailed discussion about that such as the import or export standards of waste plastics of different countries. And this issue should be considered in future studies.

We appreciate the reviewer's recognition of the difficulty in obtaining detailed information on the grade of imported waste plastics from HS trade data. However, we believe that the *RRR* data can speak for themselves. If a country imports low-quality plastic waste, this is generally reflected in lower trade values retrieved from the UN Comtrade database, leading to a lower *RRR* when other costs and benefits are held constant. Conversely, higher import costs typically indicate higher-quality waste, often seen in developed countries with a higher *RRR*. While the current HS system doesn't explicitly distinguish the quality of plastic waste, the trade value within each transaction offers an implicit measure of quality that is ultimately captured in the *RRR*. Future studies should indeed consider the variations in quality more explicitly, but the *RRR* already provides a useful approximation.

6. Line 376-381: I did not find the specific costs or cost ranks of waste plastic recycling from the references 19, 20, and 40, which makes me cannot confirm whether the top three costs are electricity, labor, and rent. Please provide the specific positions of costs in these references. Or recheck these references.

Please refer to Figure 3 in the original reference 20 (Technical, Economic, and Environmental Comparison of Closed-Loop Recycling Technologies for Common Plastics). We have attached the Figure 3 below:

In this figure, under the 'MSP' (minimum selling price) column for a functional unit of 1 kg of recycle, the top three costs, excluding the feedstock cost (light blue), are consistently electricity, capital, and labor (specifically mentioned for LDPE). We acknowledge the ambiguity and have rechecked the references to ensure accuracy.

7. Line 237-244: I think another important application of RRR is to help one country to formulate the recycling targets of waste plastic, which should be added in the main text. And RRR cannot reflect the environmental impacts of waste plastic trade.

Thank you for the insightful comment. We agree that RRR can be a valuable tool for setting recycling targets, and we've expanded on this in the revised text (lines 190 to 201). We carefully revised the sentence discussing the relationship between RRR and the environmental impacts of waste plastic trade to ensure clarity and precision.

‘*RRR* enhances the accuracy of modelling the fate and impacts of traded plastic waste, which is crucial for scientific research and policy implementation. By indicating the share of recycling versus non-recycling, *RRR* provides valuable data for measuring the environmental impacts of the global plastic waste trade, particularly in waste-importing countries. Moreover, annual *RRR* data across countries and plastic waste types sheds light on how external events influence the global plastic waste trade. For instance, a notable increase in *RRR* across many countries in 2020 coincided with a drop in crude oil prices²⁸, suggesting that lower prices for virgin and recycled plastics necessitated a higher *RRR* to cover costs and achieve profitability. In terms of policy implications, *RRR* can assist waste-importing countries in formulating and adjusting their recycling targets. Instead of relying on domestic average recycling rates, which are often based solely on domestically generated plastic waste, countries should consider separate targets for imported plastic waste, recognizing their distinct characteristics.’

8. Line 243-244: I did not understand why an increased oil price will promote an increase in RRR although you have given a short explanation.

Please refer to the revised paragraph provided above. We appreciate the reviewer pointing out the ambiguity, which helped us rephrase and clarify the explanation.